# How Far Have We Developed Antibody–Drug Conjugate for the Treatment of Cancer?

Yu Jun Lim [1], Pei Sze Clarissa Lau [1], Shi Xuan Low [1], Shong Li Ng [1], Min Yee Ong [1], Huey Ming Pang [1], Zheng Yang Lee [1], Hui Yin Yow [2], Sharina Binti Hamzah [1], Renukha Sellappans [1] and Jhi Biau Foo [1,*]

[1] School of Pharmacy, Faculty of Health & Medical Sciences, Taylor's University, 1, Jalan Taylors, Subang Jaya 47500, Malaysia; zhengyang.lee03@sd.taylors.edu.my (Z.Y.L.); sharina.hamzah@taylors.edu.my (S.B.H.); renukha.sellappans@taylors.edu.my (R.S.)

[2] Department of Pharmaceutical Life Sciences, Faculty of Pharmacy, Universiti Malaya, Jln Profesor Diraja Ungku Aziz, Seksyen 13, Kuala Lumpur 50603, Malaysia; huiyin.yow@um.edu.my

\* Correspondence: jhibiau.foo@taylors.edu.my

**Abstract:** Cancer, also known as malignant tumour or neoplasm, is a leading cause of death worldwide. One distinct feature from normal cells is that cancerous cells often overexpress protein on the cell membrane—for instance, the overexpression of human epidermal growth factor receptor 2. The expression of a specific protein on the cancerous cell surface acts as a marker that differentiates the normal cell and facilitates the recognition of cancerous cells. An emerging anticancer treatment, Antibody–Drug Conjugates (ADCs), utilises this unique feature to kill cancerous cells. ADCs consist of an antibody linked with a cytotoxic payload, mainly targeting the antigen found on cancerous cells. This design can increase the specificity in delivering the cytotoxin to the drug target, thus increasing the drug efficacy and reducing the side effect of cancer treatment due to off-target toxicities. There are tremendous quantities of clinical trials conducted to evaluate the safety and effectiveness of this magic drug in treating different types of cancers. However, only 12 ADCs have been approved by the FDA until now. This review provides the principles of ADCs and highlights the ADCs that FDA has approved. In addition, some of the ADCs that undergo clinical trials are discussed in this review. The application of computational techniques in addressing ADCs' challenges and neoantigen-targeted cancer vaccines is also highlighted. Although ADCs have been seen as promising magic drugs in cancer treatment, the problems such as toxicity, the stability of the linker, the specificity of an antibody with antigen, and so on, remain a challenge in developing ADCs.

**Keywords:** cancer; antibody–drug conjugate; clinical trials; linker; cytotoxin





## 1. Introduction

The discovery of the "Magic Bullet Theory" by a German Scientist, Paul Ehrlich, marked the targeted therapy revolution. He espoused the theory to describe his vision, where a chemical binds selectively to targeted microorganisms [1]. Ehrlich eventually realised this concept by successfully treating syphilis with a highly selective drug towards bacteria that does not harm the neighbouring healthy cells [2]. His outstanding achievement has had a significant positive impact on the healthcare system in which scientists develop drugs that could specifically target the cells in the body system.

The concept of the "magic bullet" has to some extent been realised by the development of antibody–drug conjugates (ADCs), particularly in the field of oncology. ADCs' historical revolution has answered the crucial pharmaceutical concern of all oncologists, the need for anticancer treatment to target tumour cells efficiently along with high precision and specificity. The tremendous advancement of monoclonal antibody technology has allowed various cytotoxic agents such as cytotoxic drugs, immunotoxins, and radiopharmaceuticals to be conjugated to the bullet, resulting in high selectivity and targetability on the specific antigen expressed on cancerous cells. ADCs were proposed to reduce non-specific toxicities

by altering their signalling pathways towards a therapeutic outcome or directing naturally acquired immune responses towards the tumour cells [3]. In recent years, new ADCs have been under clinical development, marking the success of this highly potential therapeutic agent in chemotherapy (Figure 1) [4]. Drago et al. (2021) have been very insightful in ADC design and maximising the potential of ADCs for cancer therapy [5]. Some of the recent reviews have also provided insightful knowledge in these areas [6–9]. Thus, in this review, the principles and challenges of ADC will be discussed, emphasising the latest development of ADCs in clinical trials. The computational techniques for addressing ADC challenges are also discussed. In addition, neoantigen-targeted cancer vaccines are also introduced to highlight the next potential area that can be used to target cancer.

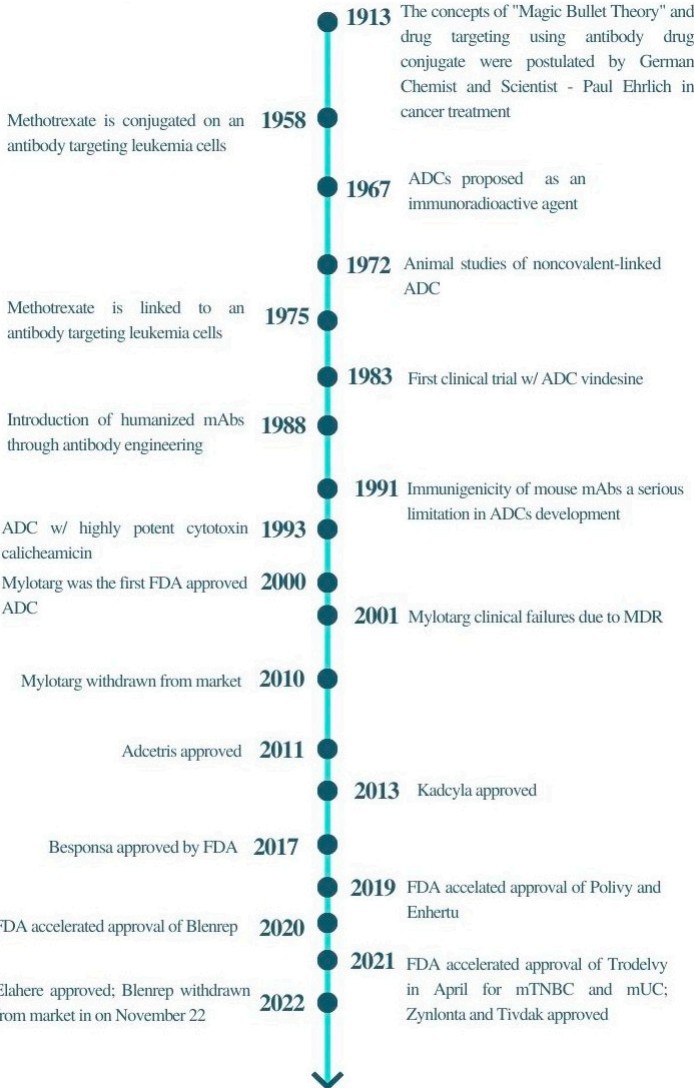

**Figure 1.** A brief timeline of ADCs development.

## 2. Principle of Antibody–Drug Conjugate

ADC comprises a tumour antigen-specific antibody, a potent cytotoxic drug and a stable chemical linker that joins the antibody to the cytotoxic drug [10]. The physicochemical properties of respective components in an ADC are summarised in Table 1.

### 2.1. Antibody

An antibody is the most crucial element in an ADC due to its role in recognising the cancer surface marker. Therefore, an excellent antibody used in ADC should be stable

during circulation in the bloodstream, less immunogenic, highly specific for the antigen, and can be internalised by the cancer cells. There are two antigen-binding fragments known as Fabs to recognise antigens expressed on cancer cells over healthy cells. A constant fragment, Fc mediates the interaction of antibodies with immune cells such as lymphocytes and B cells (Figure 2). A binding domain on the Fc receptor binds to the neonatal Fc receptor (FcRn) that regulates serum protein half-life [11]. Selection of the appropriate antibody subtype or technological engineering of the Fc domain, antigen affinity, the immune function of ADC, and biodistribution will influence the immune function of an ADC. In addition to the intrinsic cytotoxicity of drugs, the antibody on ADC mediates immune functions that are naturally acquired in patients through the activation of complement immune effector cells and signalling components, resulting in the migration of immune effector cells to the targeted sites for tumour-killing effects.

**Table 1.** Summary of the physicochemical properties of respective components in an ADC.

| Components of ADC | | |
|---|---|---|
| **Antibody** | **Subunit** | **Function** |
| | Antigen-binding Fragment (Fab) | Mediate antigen recognition expressed on tumour cells. |
| | Constant Fragment (Fc) | Facilitate binding of Fab to immune cells. |
| **Linker** | **Type** | **Mechanism of Drug Release** |
| | Cleavable | Cleavage depends on the physiological environment.<br>• Acid labile linker is cleaved at low pH, e.g., Gemtuzumab Ozogamicin.<br>• Protease cleavable linker is cleaved through proteolysis, e.g., Brentuximab Vedotin.<br>• Disulphide linker is cleaved by high intracellular glutathione concentrations, e.g., Mirvetuximab Soravtansine. |
| | Non-cleavable | It depends on lysosomal proteolytic degradation and requires optimal trafficking to lysosome. Thioether linker is cleaved by proteases in the cytosolic milieu, e.g., Trastuzumab Emtansine (T–DM1). |
| **Cytotoxic Payload** | **Target Site** | **Examples of Payload with Mechanism of Action** |
| | DNA | • Calicheamicins induce double-strand DNA breaks.<br>• Duocarmycin and Pyrrolobenzodiazepines (PBD) induce DNA alkylation by binding to A–T rich regions and guanine residues, respectively. |
| | Tubulin | • Auristatins and derivatives of Maytansine inhibit polymerisation of the microtubule and cell cycle arrest at the $G_2/M$ phase.<br>• Auristatin monomethyl auristatin E (MMAE) in Brentuximab vedotin and glembatumumab vedotin, Monomethyl auristatin F (MMAF) in Depatuximab mafodotin, a derivative of Maytansine (DM1) in trastuzumab emtansine. |

The IgG type of immunoglobulin is the most common subtype in the systemic circulation. Among the subclasses of IgG1 to IgG4, IgG1 is most frequently used due to its excellent serum stability for up to 21 days (half-life) and a higher affinity in activating Fc receptors for IgG (FcyR) fragments found on immune cells such as the natural killer cells, monocytes and macrophages. Hence, IgG1 is superior in engaging the immune system, activating Antibody-Dependent Cell-Mediated Phagocytosis (ADCP), Complement-Dependent Cytotoxicity (CDC), and Antibody-Dependent Cell-Mediated Cytotoxicity (ADCC) of cancer cell killing mechanisms [11].

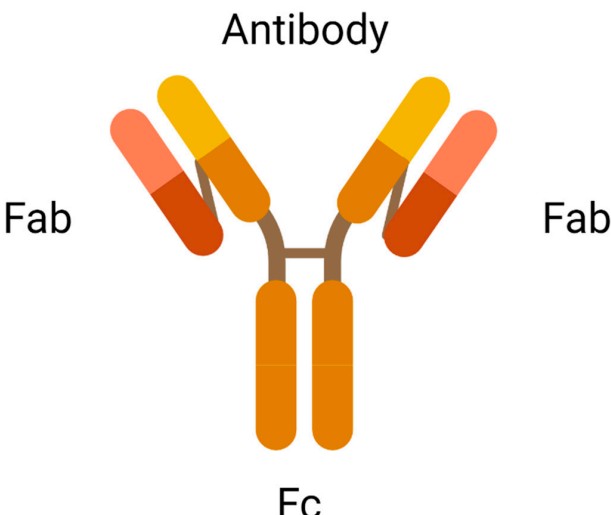

**Figure 2.** Antigen-binding fragments of an antibody (created with https://www.biorender.com/) (accessed on 2 December 2022).

### 2.2. Cytotoxic Payload

Cytotoxic payload is the effector component in ADC. The ideal candidate of an anticancer drug used in ADC should be highly potent with half-maximal inhibitory concentration (IC50) in the subnanomolar range to achieve desired therapeutic activities. Subnanomolar potency is crucial because only a few payloads can be conjugated to the targeting antibodies. Low potency of payloads results in a treatment failure. Due to the criteria of subnanomolar potency, many highly toxic drugs that were abandoned in the laboratory have now been reviewed as potential payloads in the development of ADCs. Common intracellular targets of ADC in tumour cells are DNA and Tubulin. The inhibition of these targets blocks the mitotic phase of tumour cells, slowing cell growth, and ultimately inducing apoptotic cell death [12].

Duocarmycins and Pyrrolobenzodiazepine (PDB) cause DNA alkylation by binding specifically to AT-rich regions and guanine residues, respectively. For example, the intracellular target of Maytansines and Auristatins is the tubulin found in the centrosomes. Auristatins Monomethyl Auristatin E (MMAE) and Monomethyl Auristatin F (MMAF) inhibit the polymerisation of the microtubule, thereby causing cell-cycle arrest at the G2/M phase [13]. MMAE is the effector in Brentuximab Vedotin and Glembatumumab Vedotin, while MMAF is the cytotoxic payload in Depatuxizumab Mafodotin. A highly potent maytansinoid [2].

### 2.3. Linker

The chemical linker in ADC conjoins cytotoxic payload to the antibody, stabilising the circulating ADC in the bloodstream upon administration. The functional group of linkers and the conjugation site determine the performance of ADCs in terms of circulating half-life, pharmacokinetics and pharmacodynamic profiles, and therapeutic window [14]. Disulfide, thioether, and hydrazone functional groups combine antibodies to the cytotoxic agent through intermolecular interactions. Currently, available linkers based on mechanisms of payload release are classified as cleavable or non-cleavable. Cleavable linkers depend on the physiological environment in the systemic circulation, such as pH or proteases available to release cytotoxic payload from the carrier. For example, the acid–labile linker in Gemtuzumab Ozogamicin is cleaved under low pH conditions. In contrast, protease cleavable linker in Brentuximab Vedotin is cleaved by proteases such as Cathepsin-B and plasmin. The cleavage of disulfide linkers in mirvetuximab soravtansine depends on high glutathione concentrations [15]. Non-cleavable linkers form non-reducible covalent bonds with amino acids on the antibodies. Unlike cleavable linkers, non-cleavable linkers are not

affected by the physiological environment and are more stable in blood circulation [16]. In addition, these linkers depend on complete lysosomal degradation of the mAb for payload release, which necessitates an efficient internalisation process and optimal trafficking to lysosomes—for instance, thioether linker in T–DM1 [2]. This may be an advantage since it could lead to a lower risk of systemic toxicity. One of the popular non-cleavable linkers is the polyethylene glycol (PEG) linker due to its excellent water solubility, lack of toxicity and low immunogenicity.

The average number of drug molecules conjugated to a mAb, known as the drug-to-antibody ratio (DAR), determines the potency and toxicity of ADC. Conjugation occurs on the backbone of mAb through lysine side chains on the surface of mAb or cysteine residues in the hinge regions to make interchain disulphide bonds, resulting in high variability DAR [17]. High DAR increases therapeutic efficacy and unexpectedly also increases off-target effects. Moreover, higher DAR also increases drug clearance, reducing the circulating half-life of ADC.

### 2.4. Mechanism of Action of ADC

Upon intravenous administration, ADC circulates in the bloodstream and binds to an antigen explicitly expressed by targeted tumour cells (Figure 3) [18]. Following inter-nalisation, ATP-dependent proton pumps in lysosomes to create an acidic environment to facilitate the degradation of cleavable ADCs by proteases such as plasmin and cathepsin-B, releasing cytotoxic drugs from ADC complexes [11]. In addition, lysosomal degradation, proteolytic cleavage of acid–labile linker, protease cleavable linker, and disulfide linker by the physiological environment occurs in the early or late endosomes [19]. The released cytotoxic drugs in the free circulating state resulted in the binding to intracellular targets, causing apoptosis via DNA intercalation or inhibition of microtubule polymerisation. As tumour cells lyse, free cytotoxic drugs are released and progress to bystander killing of neighbouring cancerous and non-cancerous cells via passive diffusion [20]. Apart from direct cytotoxicity, ADC-mediated effector functions include activating complement systems and infiltration of immune cells to localise at the targeted tumour via several mechanisms such as ADCP and ADCC [21].

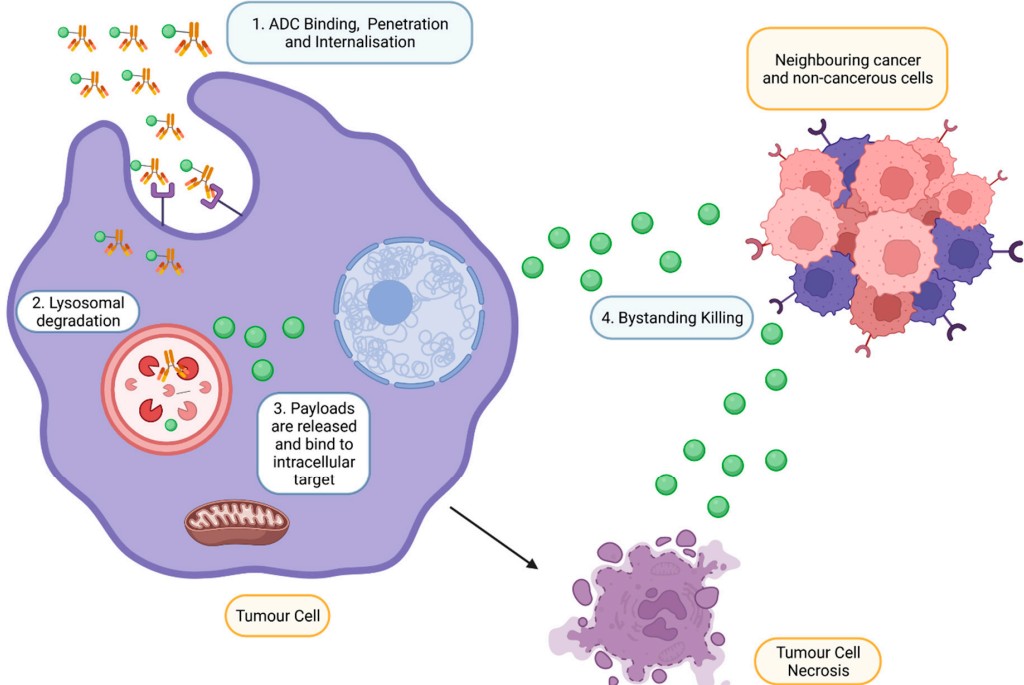

**Figure 3.** Mechanism of action of ADCs at targeted tumour cells. ADCs are rapidly internalised and degraded in lysosomes upon binding to the antigen on the tumour. Payloads are then released into

the cytoplasm, bind to the intracellular target and induce cell death. However, payloads can also exhibit bystander effects by diffusing out from the cells into adjacent cells (Created with https://www.biorender.com/) (accessed on 2 December 2022).

## 3. Antibody–Drug Conjugate Approved by FDA

Developing an ADC is not easy as the process involves complicated bioengineering, chemistry and pharmacology. Many ADCs have recently received FDA-accelerated approval to treat severe malignancies in the hospital setting despite the challenges and hurdles (Figure 4). The FDA-approved ADCs are highlighted and updated accordingly in Table 2.

(A) Ado-trastuzumab emtansine

(B) Belantamab mafodotin

**mcMMAF**

Where n ~4 mcMMAF per mAb

(C) Brentuximab vedotin

**Figure 4.** *Cont.*

(D) Enfortumab vedotin

(E) Fam-trastuzumab deruxtecan

(F) Gemtuzumab ozogamicin

**Figure 4.** *Cont.*

(G) Inotuzumab ozogamicin

(H) Polatuzumab vedotin

(**I**) Sacituzumab govitecan

**Figure 4.** *Cont.*

(J) Tisotumab vedotin

(K) Loncastuximab tesirine

**Figure 4.** Structures of (**A**) ado-trastuzumab emtansine. (**B**) belantamab mafodotin. (**C**) brentuximab vedotin. (**D**) enfortumab vedotin. (**E**) fam-trastuzumab deruxtecan. (**F**) gemtuzumab ozogamicin. (**G**) inotuzumab ozogamicin (**H**) polatuzumab vedotin. (**I**) sacituzumab govitecan. (**J**) tisotumab vedotin. (**K**) loncastuximab tesirine.

**Table 2.** FDA-approved Antibody–Drug Conjugates.

| ADC | Antibody | Target Antigen | Linker | Cytotoxic Payload | Indication | Year of Approval |
|---|---|---|---|---|---|---|
| Gemtuzumab ozogamicin | Humanised IgG4 | CD33 | Acid–labile hydrazone | Calicheamicin | Relapse or refractory AML and newly diagnosed CD33$^+$ AML | Approved in 2000, withdrawn in 2010 but then reapproved in 2017 for relapsed/refractory malignancies<br><br>FDA approved for newly diagnosed CD33+ AML in $\geq$1-month paediatric patient |
| Brentuximab vedotin | Chimeric IgG1 | CD30 | Protease-cleavable dipeptide | MMAE | HL, ALCL and different subtypes of T-cell lymphomas | FDA accelerated approval in 2011<br><br>In 2018, FDA approved the treatment of previously untreated stage III-IV HL and previously untreated ALCL and other CD30$^+$ peripheral T-cell lymphomas. |

**Table 2.** *Cont.*

| ADC | Antibody | Target Antigen | Linker | Cytotoxic Payload | Indication | Year of Approval |
|---|---|---|---|---|---|---|
| Ado-trastuzumab emtansine | Humanised IgG1 | HER2 | SMCC | DM1 | Metastatic HER2+ breast cancer, previously treated with trastuzumab and a taxane and as adjuvant treatment for HER2+ early breast cancer with the residual invasive disease after neoadjuvant taxane and trastuzumab | FDA approved in 2013<br><br>FDA approved for adjuvant treatment in 2019 |
| Inotuzumab Ozogamicin | Humanised IgG4 | CD22 | Acid–labile hydrazone | Calicheamicin | Relapsed or refractory B-cell precursor ALL | FDA approved in February 2017 |
| | | | | | Monotherapy treatment of relapsed or refractory CD22-positive B-cell precursor | EMA approved in June 2017 |
| Polatuzumab vedotin | Humanised IgG1 | CD79b | Cleavable dipeptide | MMAE | Relapsed or refractory diffuse large B-cell lymphoma | FDA accelerated approval in 2019 |
| Enfortumab vedotin | Fully human IgG1 | Nectin 4 | Protease-cleavable dipeptide (Val–Cit) linker | MMAE | Locally advanced or metastatic urothelial cancer in adult patients who received prior treatment with a PD-1/L1 inhibitor and platinum-based chemotherapy in neoadjuvant/adjuvant setting | FDA accelerated approved in 2019 |
| Trastuzumab deruxtecan | Humanised IgG1 | HER2/ERB2 | Protease-cleavable tetra-peptide (Gly–Gly–Phe–Gly) linker | DXd | Unresectable locally advanced or metastatic HER2+ breast cancer, previously treated with trastuzumab and a taxane, adjuvant treatment for HER2+ early breast cancer with the residual invasive disease after neoadjuvant taxane and trastuzumab | FDA accelerated approved in 2019 |

**Table 2.** *Cont.*

| ADC | Antibody | Target Antigen | Linker | Cytotoxic Payload | Indication | Year of Approval |
|---|---|---|---|---|---|---|
| Sacituzumab govitecan | Humanised IgG1 | TROP-2 | Acid–labile ester (CL2 linker) | SN-38 | Triple-negative breast cancer, urothelial and other cancers | FDA accelerated approval in April 2020 for mTNBC |
| | | | | | | FDA regular approval for in April 2021 TNBC |
| | | | | | | FDA accelerated approval in April 2021 for mUC |
| Belantamab mafodotin | Humanised IgG1 | BCMA | Protease-resistant maleimido-hexanoic linker | MMAF | Relapsed or refractory multiple myeloma in adults who have received at least four prior therapies | FDA accelerated approval in 2020 |
| Loncastuximab tesirine- lpyl | Humanised IgG1 | CD19 | Valine–alanine dipeptide | PDB dimer | Relapsed or refractory large B-cell lymphoma | FDA accelerated approval in April 2021 |
| Tisotumab vedotin | IgG1 | Tissue factor (TF) | mc–val–cit–PABC | MMAE | Recurrent or metastatic cervical cancer in patients with disease progression during or following chemotherapy | FDA accelerated approval in September 2021 |
| Moxetumomab Pasudotox | - | CD22 | Recombinant covalently fused | Pseudotox | Relapsed or refractory HCL who received at least two prior systemic therapies | FDA approval in September 2018 |

ADC: Antibody Drug Conjugate; AML: Acute myeloid leukaemia; MMAE: Monomethyl auristatin E; HL: Hodgkin lymphoma; ALCL: Anaplastic large cell lymphoma; ALL: Acute lymphoblastic leukaemia; mTNBC: metastatic triple-negative breast cancer; TNBC: metastatic triple-negative breast cancer; mUC: Metastatic Urothelial Carcinoma; PDB: Pyrrolobenzodiazepine; HCL: Hairy Cell Leukoma.

### 3.1. Gemtuzumab Ozogamicin

In 2000, gemtuzumab ozogamicin (GO; Mylotarg®, Wyeth Pharmaceuticals: Philadelphia, PA, USA) became the first FDA-approved ADC. It is a humanised anti-CD33 IgG4 monoclonal antibody (mAb) linked to calicheamicin through a cleavable hydrazone linker [4-(4-acetylphenoxy) butanoic acid]. In each IgG4, two or three calicheamicin molecules are bonded to it. Upon binding to the CD33 antigen, the GO–CD33 complex is internalised and fused with a lysosome. The calicheamicin derivative is released inside lysosomes, followed by DNA binding, inducing DNA double-strand break and cell death [22].

As CD33 is highly expressed primarily on myeloid cells, therefore, GO was approved to treat relapsed or refractory CD33-positive AML in adults either as monotherapy or in combination with three-cycle chemotherapy and combination with standard five-cycle chemotherapy for children aged at least two years, and monotherapy therapy of newly diagnosed CD33-positive AML in paediatric patients aged at least one month and adults. The initial accelerated FDA approval in 2000 was based on the results of three phase II trials (elderly who aged at least 60 years old with CD33-positive acute myeloid leukaemia (AML) but cannot tolerate other cytotoxic chemotherapy). Despite having a good overall response rate (ORR) (26–30%), severe adverse reactions such as hepatotoxicity, infusion responses and pulmonary toxicity were reported. GO was then withdrawn from the market in 2010 due to the reported severe side effects such as hepatic veno-occlusive disease and

delayed hematopoietic recovery [23]. The main cause of the severe side effects was the premature release of GO at a fatally toxic dose (9 mg/m$^2$) into patients' bloodstreams upon administration. This issue was then critically addressed, and another phase II clinical trial (NCT00658814) was conducted with a lower GO dosage (3 mg/m$^2$) for three doses. This trial evaluated the efficacy and side effects of combining azacitidine and GO in treating older patients (at least 60 years old) with previously untreated AML. Patients were assigned into two groups: good-risk patients (from 60 to 69 years old) or poor-risk patients (at least 70 years old). Forty-two per cent (42%) (35 of 83) of patients in the good-risk group obtained complete responses. Median relapse-free survival (RFS) and OS were reported as 8 and 11 months, respectively. Within 30 days after registration of the study, six patients died. On the other hand, only 19 of 59 participants (32%) in the poor-risk group were able to achieve CR with lower RFS (7 months) and OS (11 months), respectively. It was noted that severe toxicities were not reported in this trial compared to the earlier reported trials. Due to the positive results with lower toxicity, GO was then reapproved by FDA in 2017.

In the search for ADCs with better safety profiles and Chemistry, Manufacturing and Control (CMC), second-generation ADC drugs such as BV, T–DM1 and INO were discovered. In addition, there are a few ongoing phase II clinical trials such as (NCT02221310) to investigate the efficacy of GO in combination with chemotherapy followed by allogeneic stem cell transplantation in high-risk CD33$^+$ AML or MDS patients. The results of this trial have yet to be concluded [24]. Another phase II trial (NCT03672539) aims to determine the side effects and efficacy of liposome-encapsulated daunorubicin-cytarabine + GO in treating relapsed/refractory AML or MDS patients. This trial will be completed by the fourth quarter of 2022 [25].

### 3.2. Brentuximab Vedotin

Brentuximab vedotin (BV; Adcetris®, Seattle Genetics, Bothell, WA, USA), the second FDA-approved ADC [26], is synthesised by coupling an anti-CD30 Ab and MMAE via a protease-cleavable dipeptide linker [27]. It contains a chimeric mAb cAC10 specific for CD30, a valine–citrulline dipeptide linker and potent cytotoxic anti-tubulin agent MMAE. Each mAb molecule carries four MMAE groups. BV binds to the CD30 receptor and undergoes internalisation by endocytosis. MMAE molecules are then released into the intracellular cell when exposed to lysosomal enzymes. The microtubule network in the cell is destroyed when MMAE binds to tubulin and eventually causes cell cycle arrest in the G2/M phase and apoptosis [28].

BV is used to treat a variety of CD30-expressing lymphomas, primarily classical Hodgkins Lymphoma (cHL), in adults with a high probability of relapse and progression or failure of autologous hematopoietic stem cell transplantation (auto-HSCT). With chemotherapy, it can also treat stage III or IV cHL. Furthermore, BV can also be used to treat patients with systemic anaplastic large cell lymphoma who have failed at least one prior multi-agent chemotherapy regimen and primary cutaneous anaplastic large cell lymphoma (pcALCL) or CD30-expressing mycosis fungoides (MF) [22].

BV received accelerated approval for relapsed HL and sALCL in 2011. The approval was built on a Phase 3 study of BV in patients with a high chance of developing residual HL after hematopoietic stem-cell transplantation (HSCT) (NCT01100502). Early consolidation post autologous stem-cell transplantation (ASCT) with BV significantly improved progression-free survival (PFS) compared to placebo (median PFS 43 versus 24 months, *p* = 0.001). Neutropenia, peripheral sensory neuropathy, peripheral motor neuropathy, thrombocytopenia, and anaemia were the most prevalent treatment-emergent grade 3 adverse effects (Aes). Treatment was terminated in 33% of patients due to Aes, compared to 6% in the control group, and 53 participants died throughout the research (28 from the BV group and 25 from the placebo group) [29]. BV is currently being investigated in three phase II clinical trials (NCT03222492 and NCT03198689) in patients with diffuse cutaneous systemic sclerosis and (NCT03007030), which investigates the efficacy of BV in treating patients with CD30 positive malignant mesothelioma which cannot be removed

by surgery [30]. The estimated completion date of these trials will be the end of 2022 or in 2023.

### 3.3. Ado-Trastuzumab Emtansine

Trastuzumab emtansine (T–DM1; Kadcyla®, Genentech: South San Francisco, CA, USA), the third approved ADC, is a humanised anti-HER2 IgG1 mAb. It is chemically attached to the antimitotic maytansinoid DM1 through a [N-succinimidyl-4-(N-maleimidomethyl) cyclohexane-1-carboxylate] linker. It interacts with the sub-domain IV of the HER2 receptor and enters the cell via receptor-mediated endocytosis. Trastuzumab emtansine is degraded by lysosomes releasing DM1. DM1 then attaches to tubulin in microtubules, inhibiting microtubule function and causing cell death. The drug is approved to treat HER2-positive metastatic breast cancer patients and as adjuvant treatment for HER2+ early breast cancer in 2013 and 2019, respectively. In the KATHERINE (NCT01772472) trial, 1486 patients were randomly assigned into two groups (743 in the T–DM1 group and 743 in the trastuzumab group). Invasive disease or mortality occurred in 91 patients in the T–DM1 group (12.2%) while 165 patients were in the trastuzumab group (22.2%). The Invasive Disease-Free Survival (IDFS) rate was significantly higher in the T–DM1 group (88.3%) than in the trastuzumab group (77.0%). Distant recurrence as the first invasive-disease event occurred in 10.5% of patients in the T–DM1 group and 15.9% of those in the trastuzumab group. The safety data reported more adverse events associated with T–DM1 than with trastuzumab alone [31].

In a phase III trial (NCT01966471), T–DM1 in combination with pertuzumab was compared to trastuzumab with pertuzumab and a taxane as adjuvant therapy in patients with HER2-positive primary invasive breast cancer. The primary outcome of IDFS was carried out in Node-Positive Subpopulation and the overall population. For only the node-positive population, IDFS in T–DM1 + pertuzumab was lower than trastuzumab + pertuzumab + taxane [92.75% (CI: 90.95 to 94.54) vs. 94.10% (CI: 92.46 to 95.73)]. Similarly, the IDFS in T–DM1 + pertuzumab was also lower than trastuzumab + pertuzumab + taxane in the overall population [93.05% (CI: 91.38 to 94.72) vs. 94.22% (CI: 92.68 to 95.76)]. Apart from that, the combination of T–DM1 and pertuzumab caused a higher mortality rate (4.82%) than trastuzumab + pertuzumab + taxane (3.67%). The major serious Aes reported in these two groups were febrile neutropenia, 3.40% in T–DM1 + pertuzumab, and 5.51% in trastuzumab + pertuzumab +taxane. Other secondary outcomes such as OS, DFS and DRFI will be posted in 2023 (10 years follow-up) [32].

In the Phase III clinical trial, (NCT00829166), T–DM1 or capecitabine and lapatinib were given to 495 and 496 individuals with relapsed or primary refractory HL, respectively. Median OS with T–DM1 was 29.9 months (95% CI from 26.3 to 34.1) vs. 25.9 months in the control group (95% CI from 22.7 to 28.3). After 24.1 months of median follow-up, 136 of 496 patients (27%) transferred from the control to the T–DM1 group. A total of 51% of patients who had been randomly allocated to T–DM1 received capecitabine, and 49% received lapatinib after the study drug discontinuation. T–DM1 caused fewer grade 3 or worse adverse events than the control treatment (48% vs. 60%). Diarrhoea was the most commonly reported grade 3 or worse Aes in the control group, followed by palmar-plantar erythrodysesthesia syndrome, and vomiting. The safety profile of trastuzumab emtansine was comparable to that previously known; the most common grade 3 or worse adverse events in the trastuzumab emtansine group are thrombocytopenia, elevated aspartate aminotransferase levels, and anaemia. Nine individuals from both treatment and control groups died from the trial, five of which were probably related to the treatment [33].

There are a few ongoing studies to evaluate the efficacy and safety of Trastuzumab Deruxtecan compared with T–DM1 in high-risk patients with residual invasive breast cancer following neoadjuvant therapy (NCT04622319) [34] and T–DM1 plus atezolizumab compared with T–DM1 plus placebo in patients with HER2-positive and PD-L1-positive LABC or MBC (NCT04740918) [35]. The results have yet to be released.

### 3.4. Inotuzumab Ozogamicin

Inotuzumab Ozogamicin (InO; Besponsa®, Wyeth Pharmaceuticals: Philadelphia, PA, USA), the fourth approved ADC drug introduced in 2017, consists of humanised mAb IgG4 and calicheamicin derivative via a cleavable hydrazone linker [27]. The complex is directed against CD22-expressing tumour cells, which are mainly expressed on the masses of B-cell lymphoblasts. It can treat relapsed or refractory (R/R) B-cell precursor acute lymphoblastic leukaemia (ALL). The antibody–antigen complex is rapidly internalised upon binding to CD22 receptors found on the leukemic cells surface, forming an endosome following fusion with lysosomes. The cytotoxic calicheamicin derivative is then released inside the cell. This leads to double-strand DNA breakage, proceeding to the arrest of the cell cycle and apoptosis. Note that the effect of calicheamicin is independent of the cell cycle progression, hence able to induce apoptosis even in those rapidly dividing cells [36].

The approval of InO was based on the INO-VATE (NCT01564784) phase III trial result [37]. All patients were treated with InO or with the investigator's standard-of-care (SoC) choice in this randomised trial. In the primary intention-to-treat analysis (ITT), 218 participants randomised were analysed for complete remission. In the InO arm, the complete remission rate was significantly higher-achieving CR/CRi of 80.7% versus 29.4% in the SoC group ($p < 0.001$). Among participants who had CR, the minimal residual disease negativity (MDR) rate was higher in the InO group (78.4% vs. 28.1%, $p < 0.001$), in which a larger number of patients successfully proceeded to HSCT (41% vs. 11%). Similarly, the duration of remission (DoR) in the InO arm was longer, with a median of 4.6 months compared with the comparator's arm, with a median of 3.1 months. Secondly, the survival threshold was analysed where the PFS of 5.0 months in the InO group was significantly higher than only 1.8 months in the SoC group ($p < 0.001$). Correspondingly, the median OS was 7.7 months as opposed to 6.7 months, with a hazard ratio of 0.77. In the safety analysis, AEs related to haematology were at a higher rate at all-grade, grade 3 and higher events in the InO group. All-grade hepatotoxicity of veno-occlusive liver disease (VOS/SOS) was higher in the InO arm (11%) versus the SoC group (1%) [38]. Other reported haematologic grade 3 adverse events included neutropenia, thrombocytopenia, leukopenia, and febrile neutropenia.

In a small study of 18 participants recruited in determining the safety of InO in combination with bosutinib, 83% CR/CRi and 61% negative MDR were achieved. Upon the subsequent 44 months follow-up, the median DoR and OS were 7.7 months and 13.5 months, respectively. Six patients proceeded to HSCT. None of the patients experienced VOS/SOS. The treatment efficacy of Inotuzumab ozogamicin in combination with bosutinib was well tolerated in the R/R Philadelphia chromosome-positive ALL and lymphoid blast phase of chronic myeloid leukaemia [39]. The current ongoing clinical trials evaluate the use of either monotherapy or combination therapy in the R/R or upfront setting. Examples of efficacy evaluation trials include InO treatment of younger patients with R/R B-cell ALL (NCT02981628), maintenance therapy with InO in post-transplant (NCT03104491) and addition of InO to front-line setting with chemotherapy in young adults (NCT03150693) [36]. Compared to other current ADCs in the market, the ongoing clinical trials in evaluating InO are widely analysed and studied due to outstanding efficacy with a high complete remission rate of the drug in the previous clinical trials.

### 3.5. Polatuzumab Vedotin-Piiq

Polatuzumab vedotin (Polivy™, Genentech: South San Francisco, United States) consists of humanised IgG1 antibody targeted to CD79b, conjugated to the antineoplastic agent (MMAE) via a protease-cleavable dipeptide linker [27]. Upon CD79 binding to the B-cell surface, internalisation and cleavage occur, allowing the release of MMAE into the cell. MMAE then binds to tubules, disturbing the network of cellular microtubules and inhibiting tubulin polymerisation, thus inducing cell cycle arrest. It is indicated to treat R/R diffuse large B-cell lymphoma (DLBCL) in adults who have progressed after ≥2 therapies with bendamustine and rituximab (BR) [40].

The combination of Pola–BR was granted accelerated approval in June 2019 according to the phase Ib/II GO29365 trial (NCT02257567). The clinical profile of Pola–BR was compared with BR. Regarding the complete remission rate, the combination therapy of Pola–BR was significantly more favourable than BR (40% vs. 18%) ($p$ = 0.026). At the end of treatment, Pola–BR's OS was 45% compared to 18% in BR. Correspondingly, the DoR was longer with Pola–BR of 10.3 months versus BR of 4.1 months. Median PFS was significantly higher in Pola–BR (7.6 vs. 2.0 months; $p$ < 0.0001) [40]. The ORR was higher in the combination regimen than in BR alone (45% vs. 18%) [27]. Pola–BR treated patients experienced higher grade 3–4 adverse events, including neutropenia (46.2% vs. 33.3%), thrombocytopenia (41% vs. 23.1%), and anaemia (28.2% vs. 17.9%). Nevertheless, grade 3–4 infestations and infections were similar in both groups (23.1% vs. 20.5%) [41]. There are lower fatal adverse events in the Pola–BR arm than in the BR arm (9 vs. 11 patients), in which the common cause was infection [42]. The extended follow-up (median 30 months) data demonstrated that the clinical benefits are sustained in the Pola–BR arm. The PFS of INV-assessed in the Pola–BR arm was 7.5 months compared to 2.0 months in the BR arm ($p$ < 0.0001). Likewise, median OS and median DoR were higher in the Pola–BR treated group. In patients with a confirmed positive response to Pola–BR, nearly half of them remained event-free (47%). No odd safety signals were observed in the extended follow-up [42].

Current ongoing clinical trials are the phase III trial evaluating the clinical profile of combination therapy of Polatuzumab Vedotin with rituximab–CHP in comparison to rituximab–CHOP in earlier untreated DLBCL patients (NCT03274492; POLARIX), phase 1b/2 trials evaluating the efficacy and safety of polatuzumab vedotin and immunochemotherapy in patients with previously untreated DLBCL (rituximab–CHP or mosunetuzumab–CHP; NCT03677141), or in patients with R/R follicular- or DLBCL (bendamustine with either obinutuzumab or rituximab; NCT02257567). Another combination therapy with rituximab or obinutuzumab is also being evaluated with an immunomodulating agent, lenalidomide, in R/R DLBCL or FL setting (NCT02600897) [40].

*3.6. Enfortumab Vedotin*

Enfortumab vedotin (EV, Padcev®, Astellas Pharma US: Northbrook, United States) is the first Nectin 4-targeting ADC that was approved to treat patients with locally advanced or metastatic urothelial cancer (Ia/mUC) who have previously received anti-PD-1/PD-L1 inhibitors and platinum-based chemotherapy (CTX) in neoadjuvant/adjuvant settings. Nectin 4-directed IgG1 mAb is coupled to the MMAE through an mc–Val–Cit–PABC linker to form an EV. Its antibody component attaches to the Nectin-4 protein on bladder cancer cells. The MMAE is released when EV penetrates the cell, inducing cell death. Nectin 4 was found highly expressed in bladder, pancreatic, breast, and lung cancers but low-to-moderately expressed in normal epithelia [43].

EV received accelerated approval during the ongoing phase II clinical trial (NCT03219333) [44]. It was given to 125 patients with metastatic UC, progressive under platinum-based CTX and immune checkpoint inhibition (ICPI). The ORR of 44% was reported (95% CI: 35.1% to 53.2%), with 12% of CRR. Prespecified subgroups, including those with liver metastases and previously failed anti-PD-1/L1 therapy participants, showed similar results. The median PFS and OS were 5.8 and 11.7 months, respectively. Although severe toxicities were rare, rash and fatigue were the most prevalent grade 1–2 treatment-related AEs. Due to peripheral sensory neuropathy, most of the patients' treatment doses were adjusted or terminated. These findings prompted dozens of new researches to explore EV use for UC in various circumstances. In another Phase III clinical trial (NCT03474107), 608 participants were randomised, with 301 receiving EV and 307 receiving CTX [45]. Three hundred and one (301) deaths had occurred during the study (134 from the EV group and 167 from the CTX group). The EV group had a better OS rate than the CTX group (12.88 vs. 8.97 months, $p$ = 0.001). It also had a better PFS rate than the CTX group (5.55 versus. 3.71 months, $p$ < 0.001). The Aes rate in both groups was similar (93.9% and 91.8% in EV and CTX groups,

respectively); the Aes rate of ≥grade 3 was also similar (51.4% and 49.8%, respectively) [46]. Another trial to investigate the efficacy and safety of EV in the treatment of Non-muscle Invasive Bladder Cancer (NMIBC) patients (NCT05014139) is also initiated. It is hoped that positive outcomes will be reported soon.

### 3.7. Fam-Trastuzumab Deruxtecan

Fam–trastuzumab deruxtecan (Enhertu®, Daiichi Sankyo, DS-8201a, Luitpoldstraße, Germany) is an ADC that combines an anti-HER2 humanised Ab and a potent topoisomerase I inhibitor through a glycine–glycine–phenylalanine–glycine linker. The payload release will damage cancerous cells' DNA and induce apoptotic cell death. Enhertu® received accelerated approval in 2019 and thus became the first site-specific ADC. Based on the phase I/II trial result (NCT03248492), participants with HER2+ breast cancer who could not tolerate T–DM1 due to toxicities or other circumstances were randomised into three different dose groups in the first stage of the study. In the second part, 184 patients with a median of six previous lines received the 5.4 mg/kg dose established in the first section [47]. T–DM1 and trastuzumab (100%), pertuzumab (65.8%), additional anti-HER2 drugs (54.3%), hormone treatments (48.9%), and other systemic therapies (48.9%) were among the previous therapy lines (99.5%). Overall, the outcomes from the major subgroups had consistency; the reported ORR was 60.9%, while the DCR and clinical benefit rates were 97.3% and 76.1%, respectively. The OS rate was documented at 6 and 12 months, with 93.9% and 86.2%, respectively, while the median OS was not achieved when the study was published. From this study, 20% of patients experienced significant treatment-emergent AEs [22]. Gastrointestinal and haematological problems were the most common AEs. Treatment in 9% of patients was terminated due to interstitial lung disease. The mortality rate due to treatment was 4.3%, with pulmonary toxicity or interstitial lung disease being the most common cause of death. The ongoing clinical trials (NCT04739761) [48] are accessing T–DM1 in individuals with or without brain metastasis who have previously been treated for advanced or metastatic HER2 breast cancer and HER2 mutated Metastatic Non-small-cell Lung Cancer (NSCLC) (NCT04644237).

### 3.8. Sacituzumab Govitecan

Sacituzumab Govitecan (Trodelvy™, Gilead Sciences, Foster City, CA, USA) (IMMU-132) consists of a humanised IgG mAb that binds to trophoblast cell surface antigen 2 (TROP-2), which is conjugated with SN-38 (a topoisomerase-I inhibitor) via CL2 acid-liable linker [27]. During internalisation, the premature release of SN-38 before the intact conjugate internalisation (bystander effect) might affect the therapeutic activity of IMMU-132. Nevertheless, SN-38 displays a markedly diverge mechanism since it is released slowly and locally compared to others ADCs. As SN-38 is inhibiting the S-phase of the cell cycle, the maintenance of a low concentration of SN-38 in tumours over a period may enhance the potency, in accordance with the repeated high dosage of drug administration [49].

According to the FDA (2021), sacituzumab govitecan was granted accelerated approval in April 2020 for metastatic TNBC with ≥2 prior therapies. The confirmatory trial was supported by a randomised trial (ASCENT, NCT02574455) [50]. Randomisation was conducted with sacituzumab govitecan 10 mg/kg IV infusion (n = 267) versus single-agent chemotherapy of the physician's choice (n = 262). The primary efficacy endpoint was median PFS where it was longer in the sacituzumab govitecan arm versus the comparator arm (4.8 vs. 1.7 months). In addition, the median OS was 11.8 months versus 6.9 months, respectively. On 7 April 2021, sacituzumab govitecan received regular approval for patients with locally unresectable advanced- or mTNBC who have undergone ≥ 2 prior systemic therapies, with at least one for metastatic disease. The most common adverse reactions observed (>25%) in sacituzumab govitecan treated patients include neutropenia, diarrhoea, nausea, alopecia, fatigue and vomiting, which was considered a very acceptable safety profile. Another study of single-agent administered sacituzumab govitecan in adults with epithelial cancer (NCT01631552) [51] was conducted. In comparing patients'

response rate with metastatic, platinum-resistant urothelial carcinoma under second-line chemotherapy (<20%), with a median OS of <1 year, sacituzumab govitecan treatment provided a significant clinical response on PFS and OS with 6.7–8.2, and 7.5–11.4 months, respectively. Following that, a phase II trial, a novel therapeutic choice for treating PRUC, commenced [52]. On 13 April 2021, sacituzumab govitecan received accelerated approval for patients with locally advanced- or metastatic UC who had preceded with platinum-regimens chemotherapy. Sacituzumab govitecan was also studied with programmed death receptor-1/death-ligand-1 (PD-1/PD-L1) inhibitor (NCT03547973) [53]. The primary end-points of efficacy observed were ORR and DoR with 27.5% and 7.2 months, respectively. The recorded complete response was 5.4% and a 22.3% partial response. Examples of current ongoing clinical trials include a phase III study of HR+/HER2–MBC patient who is not responding to ≥2 prior chemotherapy treatment (NCT03901339) [54] and a phase II study in solid metastatic tumours (NCT03964727) [55].

### 3.9. Belantamab Mafodotin-Blmf

Belantamab mafodotin (BLENREP™, GlaxoSmithKline, Strada Provinciale Asolana, Italy) consists of humanised afucosylated IgG1 antibody that binds to B-cell maturation antigen (BCMA) and conjugates to microtubule inhibitor monomethyl auristatin-F (MMAF) via a protease-resistant linker maleimidohexanoic. Upon binding to BCMA, internalisation occurs rapidly, and the cys–mcMMAF cytotoxic moiety is then delivered and released to the target therapeutic site. The cys–mcMMAF interferes with the microtubule network of dividing cells, causing cell cycle arrest at the G2/M phases and apoptosis. It is indicated for the treatment of adults with R/R multiple myeloma who have received ≥ 4 prior therapies (a proteasome inhibitor, an anti-CD38 monoclonal antibody and an immunomodulatory agent) [56].

Belantamab mafodotin was given accelerated approval based on the DREAMM-2 (NCT 03525678) [57] trial that investigated the effectiveness and adverse event of 2 doses in multiple myeloma patients who failed their previous treatment with the anti-CD38 antibody. The randomised assignment was performed where one group received belantamab mafodotin intravenously at 2.5 mg/kg (n = 97) or a higher dose of 3.4 mg/kg (n = 99). A clinically significant ORR of 31% vs. 34% was achieved in phase 2. An excellent partial response (VGPR)of 19% vs. 20% was observed. A total of 73% of patients had a duration of response of more than six months. In terms of side effect profile, the common grade 3/4 adverse events were keratopathy (27% vs. 21%), thrombocytopenia (20% vs. 33%), and anaemia (20% vs. 25%). There were two deaths potentially related to treatment: sepsis in the 2.5 mg/kg group and hemophagocytic lymph histiocytosis in the 3.4 mg/kg group [58]. Potential resistance mechanisms related to belantamab mafodotin had been suggested; hence drug candidates and strategies are studied to counter them, including new drug combination regimens (NCT02064387) [59].

### 3.10. Loncastuximab Tesirine-Lpyl

Loncastuximab tesirine (Zynlonta™, ADC Therapeutics SA, Route de la Corniche, Switzerland) is a CD19-directed antibody conjugated to pyrrolobenzodiazepine (PBD) dimer via a valine–alanine cleavable maleimide linker. PBD dimers are the polymer of PBD monomers that perform cross-linkage at specific sites of DNA [60]. Hence, the division of cancer cells ceases, followed by cell death. Upon binding to the CD-19-expressing tumours, internalisation by the cell occurs, allowing loncastuximab tesirine to bind irreversibly to the DNA. As a result, the metabolism of DNA is disrupted [61].

Loncastuximab tesirine was granted accelerated approval in April 2021 to treat relapsed or refractory large B-cell lymphoma in adult patients. The criteria were patients undergoing two or more lines of systemic therapy in the treatment of diffuse large B-cell lymphoma. The approval was supported by a clinical trial evaluating the safety and anti-tumour activity of Loncastuximab Tesirine, ADCT-402. The participants were all with lymphoma diseases, including large B-Cell, Mantle Cell or Follicular Lymphoma

(NCT03685344) [62]. The overall result presented that the tolerable dose of Loncastuximab tesirine was 3 plus 3 dose-escalation at 15–200 μg/kg. The dose expansion was at 120 and 150 μg/kg. The clinical activity, pharmacokinetics and immunogenicity of Loncastuximab tesirine achieved the objective of maximum tolerance and recommended dose in the safety evaluation study. Only four patients were reported with dose-limiting hematologic toxicities. Other adverse events include fatigue, nausea, oedema, and liver enzyme abnormalities. The ORR and CR were 45.6% and 26.7%, respectively. Among the varieties of the diseases, follicular lymphoma participants showed the highest ORR at 78.6%. There are a few ongoing clinical trials with Loncastuximab as it just received accelerated approval. For example, the evaluation of combination therapy of Loncastuximab tersirine with Ibrutinib in treating lymphoma (NCT03684694) [62] and the combination therapy with rituximab compared to the standard immunochemotherapy (NCT04384484) [63].

### 3.11. Tisotumab Vedotin

Tisotumab vedotin, known as Tivdak (company: Seagen, location: Bothell, United states), is an ADC containing human monoclonal antibody that targets the tissue factor (TF), which is aberrantly expressed by numerous solid tumours. The human IgG1 monoclonal antibody links to the cytotoxin MMAE via a cleavable mc–val–cit–PABC type linker. Upon binding and internalisation of tisotumab vedotin, the linker will be cleaved, and the payload MMAE will be released, which then induces cell death by microtubule disruption [64].

Tivdak received accelerated approval from FDA on 20 September 2021 from a phase II clinical trial (NCT03438396) [65]. The trial investigated the safety, quality and efficacy of tisotumab vedotin in patients with recurrent or metastatic cervical cancer. One hundred two (102) participants were enrolled and treated with 2.0 mg/kg of Tivdak every three weeks. The outcome was supposed to be measured based on the time frame of up to 2 years. Despite achieving an ORR of 23.8% (95% CI: 15.9 to 33.3), the median response duration was only 8.3 months (95% CI: 4.2, not reached). Anaemia (32.67%), eye diseases such as dry eyes (24.75%) and keratitis (10.89%), fatigue (34.65%), GI discomfort, conjunctival infection (30.69%), and epistasis (38.61%) are among the most prevalent side effects. Tivdak also causes alopecia (38.61%), pruritus (12.87%), rash (12.87%), as well as other issues. The trial result shows that tisotumab vedotin possesses a clinically durable and meaningful antitumour activity in women with metastatic or recurrent cervical cancer, meanwhile having a tolerable and manageable safety profile [66]. The ongoing trial (NCT03485209) assesses the efficacy, safety and tolerability of tisotumab vedotin in solid tumours. The estimated completion date will be at the end of 2024. Another clinical trial (NCT03786081) investigates the safety and efficacy of Tivdak monotherapy in combination with other cancer agents such as Bevacizumab, Pembrolizumab or Carboplatin in cervical cancer patients.

### 3.12. Moxetumomab Pasudotox

Moxetumomab pasudotox (Lumoxiti, company: AstraZeneca Pharmaceuticals LP, location: Wilmington, United States) is a CD22-directed cytotoxin used to treat relapsed or refractory hairy cell leukaemia (HCL) in patients who have previously received at least two systemic treatments, including a purine nucleoside analogue. After binding the monoclonal antibody to CD22, Lumoxiti is internalised, resulting in the ADP–ribosylation of factor 2 elongation. The modified toxin will be released, inhibiting the protein translation, and inducing apoptosis in cancer cells with a high CD22 expression level.

FDA approved moxetumomab pasudotox in September 2018 based on NCT01829711 [67]. In this trial, patients received 40 μg/kg moxetumomab pasudotox intravenously on days 1, 3, and 5 of each 28-day cycle for up to six cycles. The durable CR assessed by blinded independent central review was 36.3% (95% CI: 25.8 to 47.8), while CR evaluated by the investigator was 48.8% (95% CI: 37.4 to 60.2). The Minimal Residual Disease (MRD) negative CR assessed by the blinded independent central review was 33.8% (95% CI: 23.6 to 45.2) while the investigator's assessment was slightly lower [32.5% (95% CI: 22.4 to 43.9)]. The median PFS was 41.5 months (95% CI: 28.1 to 71.7). Febrile neutropenia

and haemolytic uraemic syndrome are two significant side effects. Peripheral oedema, fatigue, diarrhoea, constipation, anaemia, pyrexia, and nausea are other AEs. The safety of moxetumomab pasudotox in combination with Rituximab or Ruxience for people with HCL or in other variations of HCL is also being investigated (NCT03805932). The results of the trial, however, are yet to be released.

## 4. Antibody–Drug Conjugates under Development

In addition to the ADCs already approved by the FDA, many ADCs are currently undergoing development and clinical trials. These ADCs use different antibodies that target specific antigens, conjugated with a cytotoxic payload, which show a precision killing effect towards the cancer cell. Some of the interesting new ADCs undergoing clinical trials are summarised in Table 3 and discussed in this section.

**Table 3.** New antibody–drug conjugates that have entered clinical trials.

| Compound Name | Target Antigen (A), Linker (L), Cytotoxin (C) | Indication | Phase | Efficacy | Side Effect | Reference |
|---|---|---|---|---|---|---|
| AGS-16C3F | A: ENPP3<br><br>L: Non-cleavable maleimido-caproyl linker<br><br>C: MMAF | Renal cell carcinoma | II | Overall Survival<br>13.1 months<br><br>Progression-Free Survival<br>2.9 months | Fatigue<br>Nausea<br>Blurred vision | NCT02639182<br><br>Status: Completed<br><br>[68] |
| AGS62P1 | A: FLT3<br><br>L: Non-cleavable linker<br><br>C: AGL-0182-30 | Acute Myeloid Leukemia | I | *N/A*<br>*Last Update Posted: 18 June 2019* | | NCT02864290<br><br>Status: Terminated (due to lack of efficacy)<br><br>[69] |
| AGS67E | A: CD37<br><br>L: Protease cleavable linker<br><br>C: MMAE | Refractory or Relapsed Lymphoid Malignancies | I | Overall Response Rate<br>22%<br><br>Complete Response Rate<br>14% | Peripheral neuropathy<br>Neutropenia | NCT02175433<br><br>Status: Completed<br><br>[70] |
| Anetumab Ravtansine (Bay 94-9343) | A: Mesothelin<br><br>L: Reducible SPDB<br><br>C: DM4 | Mesothelin-expressing Pancreatic Cancer | II | Response Rate<br>Progressive disease: 85.7%<br>Stable disease: 14.3%<br><br>Time to Progression<br>Median: 63.5 days | Anaemia<br>Cardiac disorders<br>Eye disorders | NCT03023722<br><br>Status: Completed<br><br>[71] |
| ARX788 | A: HER2<br><br>L: pAcF<br><br>C: MMAF | HER2 Positive Metastatic Breast Cancer | II | *N/A (Ongoing)*<br>*Last Update Posted: 11 February 2022*<br>*Estimated Study Completion Date: February 2025* | | NCT04829604<br><br>Status: Recruiting<br><br>[72] |

**Table 3.** *Cont.*

| Compound Name | Target Antigen (A), Linker (L), Cytotoxin (C) | Indication | Phase | Efficacy | Side Effect | Reference |
|---|---|---|---|---|---|---|
| | | Breast Neoplasms, Gastric Neoplasm and Solid tumours | I | *N/A (Ongoing)* *Last Update Posted: 2 September 2021* *Estimated Study Completion Date: March 2023* | | NCT03255070 Status: Recruiting [73] |
| BA3011 (CAB-AXL-ADC) | A: AXL L: Cleavable linker C: MMAE | Solid tumours | I/II | *N/A (Ongoing)* *Last Update Posted: 13 September 2021* *Estimated Study Completion Date: January 2022* | | NCT03425279 Status: Recruiting [74] |
| BA3021 (CAB-ROR2-ADC) | A: Ror2 L: *N/A* C: *N/A* | Solid tumours | I/II | *N/A (Ongoing)* *Last Update Posted: 13 September 2021* *Estimated Study Completion Date: 30 June 2023* | | NCT03504488 Status: Recruiting [75] |
| BMS-986148 | A: Mesothelin L: *N/A* C: Tubulysin | Advance solid tumours: mesothelioma, non-small cell lung cancer, ovarian cancer, pancreatic cancer, and gastric cancer. | I/II | Objective Response Rate Monotherapy: 6% Combo-therapy: 20% [76] | AST increased ALT increased Fatigue Nausea [76] | NCT02341625 Status: Terminated (due to business reasons not related to safety) [77] |
| Camidanlumab Tesirine (ADCT-301) | A: CD25 L: Cleavable valine–alanine C: PBD dimer | Relapsed or refractory Hodgkin's Lymphoma | II | *N/A (Ongoing)* *Last Update Posted: 5 February 2021* *Estimated Study Completion Date: 15 May 2024* | | NCT04052997 Status: Active, not-recruiting [78] |
| Coltuximab ravtansine (SAR3419) | A: CD19 L: Cleavable disulfide, C: DM4 | Diffuse large B-cell lymphoma | II | Overall Response Rate 43.9% Complete Responses: 14.6% Partial Responses: 29.3% | AstheniaFatigue Nausea Diarrhoea Cough AST increased Hematologic disorders | NCT01472887 Status: Completed [79] |
| | | Acute Lymphoblastic Leukaemia | | Objective Response Rate 25.47% Duration of Response 1.94 months | Pyrexia Diarrhoea Nausea Haematologic disorders | NCT01440179 Status: Terminated (due to very modest activity compared to competitors) [80] |

**Table 3.** *Cont.*

| Compound Name | Target Antigen (A), Linker (L), Cytotoxin (C) | Indication | Phase | Efficacy | Side Effect | Reference |
|---|---|---|---|---|---|---|
| CX-2029 | A: CD71<br><br>L: Cleavable protease<br><br>C: MMAE | Solid tumours, diffuse large B-cell lymphoma | I/II | *N/A (Ongoing)*<br>*Last Update Posted: 15 December 2021*<br>*Estimated Study Completion Date: December 2022* | | NCT03543813<br><br>Status: Active, not-recruiting<br><br>[81] |
| Datopotamab deruxtecan DS-1062a | A: TACSTD<br><br>L: Tetrapeptide<br><br>C: Topoisomerase I inhibitor | Advanced Solid Tumours | I | *N/A (Ongoing)*<br>*Last Update Posted: 26 January 2022*<br>*Estimated Study Completion Date: 1 January 2024* | | NCT03401385<br><br>Status: Recruiting<br><br>[82] |
| Denintuzumab mafodotin (SGN-CD19A) | A: CD19<br><br>L: Non-cleavable maleimidocaproxyl-valine-citrulline<br><br>C: MMAF | B-cell lymphoma | I | Complete Remission<br>Q3wk dosing relapsed pts: 7<br><br>Partial Remission<br>Q3wk dosing relapsed pts: 4<br><br>Complete Remission Rate<br>Q3wk dosing relapsed pts: 32%<br>Q3wk dosing refractory pts: 10% | Blurred vision Dry eye Fatigue Keratopathy Constipation Photophobia Nausea | NCT01786135<br><br>Status: Completed<br><br>[83] |
| | | Leukaemia and Lymphoma | | Complete Remission<br>Weekly dosing: 6<br>Q3wk dosing: 3<br><br>Partial Remission<br>Weekly dosing: 1<br><br>Complete Remission Rate<br>Weekly dosing: 19%<br>Q3wk dosing: 35% | Pyrexia Nausea Fatigue Headache Chills Vomiting Blurred vision Anaemia | NCT01786096<br><br>Status: Completed<br><br>[84] |
| Depatuxizumab mafodotin (ABT-414) | A: EGFR<br><br>L: Non-cleavable maleimido-caproyl linker<br><br>C: MMAF | Glioblastoma Multiforme | I | Objective Response Rate<br>14.3%<br><br>6-month Progression-Free Survival Rate<br>25.2%<br><br>6-month Overall Survival Rate<br>69.1%<br><br>[85] | Blurred vision Fatigue Photophobia<br><br>[85] | NCT01800695<br><br>Status: Completed<br><br>[86] |

| Compound Name | Target Antigen (A), Linker (L), Cytotoxin (C) | Indication | Phase | Efficacy | Side Effect | Reference |
|---|---|---|---|---|---|---|
| | | Glioblastoma | II | <u>Overall Survival</u><br>15.5 months<br><br><u>Progression-Free Survival</u><br>3.5 months | Keratitis<br>Fatigue<br>Dry eye<br>Headache | NCT02343406<br><br>Status: Completed<br><br>[87] |
| | | Glioblastoma and Gliosarcoma | II/III | *N/A (Ongoing)*<br>*Last Update Posted: 11 January 2022*<br>*Estimated Study Completion Date:*<br>*31 December 2021* | | NCT02573324<br><br>Status: Active, not recruiting<br><br>[88] |
| Disitamab vedotin (RC-48) | A: HER2/neu<br><br>L: Cleavable linker<br><br>C: MMAE | HER2-positive advanced malignant solid tumours | I | <u>Overall Response Rate:</u><br>21%<br><br><u>Disease Control Rate</u><br>49.1% | Neutropenia<br>Hypoesthesia<br>Increased conjugation of blood bilirubin | NCT02881190<br><br>Status: Completed<br><br>[89] |
| DS-7300a (B7-H3 ADC) | A: B7-H3 (CD276)<br><br>L: Cleavable peptide linker<br><br>C: DXd (exatecan derivative) | Advanced and malignant solid tumours | I/II | *N/A (Ongoing)*<br>*Last Update Posted: 14 December 2021*<br>*Estimated Study Completion Date: 1 October 2023* | | NCT04145622<br><br>Status: Recruiting<br><br>[90] |
| Enapotamab vedotin (HuMax-AXL-ADC) | A: AXL<br><br>L: Cleavable mc–val–cit–PABC<br><br>C: MMAE | Non-small cell lung cancer | I/II | <u>Objective Response Rate</u><br>19%<br><br><u>Disease Control Rate</u><br>50%<br><br>[91] | Constipation<br>Nausea<br>Decreased appetite<br><br>[91] | NCT02988817<br><br>Status: Completed<br><br>[92] |
| Epratuzumab tesirine (ADCT-602) | A: CD22<br><br>L: Cleavable valine–alanine peptide<br><br>C: PBD dimer | Recurrent or refractory B-cell acute lymphoblastic leukaemia | I/II | *N/A (Ongoing)*<br>*Last Update Posted: 26 January 2022*<br>*Estimated Study Completion Date:*<br>*31 December 2022* | | NCT03698552<br><br>Status: Recruiting<br><br>[25] |

**Table 3.** *Cont.*

| Compound Name | Target Antigen (A), Linker (L), Cytotoxin (C) | Indication | Phase | Efficacy | Side Effect | Reference |
|---|---|---|---|---|---|---|
| Glembatumumab vedotin (CDX-011) | A: gpNMB, L: Cleavable Dipeptide C: MMAE | Melanoma | II | Overall Survival 8.8 months Progression-Free Survival 4.4 months | Alopecia Peripheral sensory neuropathy Fatigue Nausea Decreased appetite | NCT02302339 Status: Terminated (due to development of glembatumumab vedotin was discontinued) [93] |
| Iladatuzumab vedotin (DCDS0780A) | A: CD79B L: Cleavable linker C: MMAE | Non-Hodgkin's Lymphoma | I | Response rate 47% Complete responses: 28% Partial responses: 18% [94] | Blurred vision Fatigue Corneal deposits Neutropenia Nausea Peripheral neuropathy [94] | NCT02453087 Status: Completed [95] |
| Ladiratuzumab vedotin (SGN-LIV1A) | A: Zinc transporter LIV-1 L: Protease-cleavable linker C: MMAE | Locally advanced, Metastatic triple-negative breast cancer | I/II | *N/A (Ongoing)* *Last Update Posted: 11 January 2022* *Estimated Study Completion Date: 31 December 2024* | | NCT03310957 Status: Recruiting [96] |
| Lifastuzumab vedotin (DNIB0600A) | A: Phosphate-sodium cotransporter SLC34A2 L: Cleavable maleimidocaproyl-valyl-citrullinyl-p-aminobenzyloxycarbonyl C: MMAE | Non-Small Cell Lung Cancer (NSCLC) and Platinum Resistant Ovarian Cancer | I | Duration of Response NSCLC: 5 months PROC: 11.4 months | Fatigue Nausea Decreased appetite Vomiting Peripheral sensory neuropathy | NCT01363947 Status: Completed [97] |
| Lorvotuzumab mertansine (IMGN901) | A: CD56 L: Cleavable disulfide C: DM1 | Small Cell Lung Cancer | II | Overall Survival 10.1 months Progression-Free Survival 6.2 months | Peripheral Sensory Neuropathy Anaemia Neutropenia Fatigue | NCT01237678 Status: Terminated(Study was stopped earlydue to lack of efficacy signal and safety concerns) [98] |

**Table 3.** *Cont.*

| Compound Name | Target Antigen (A), Linker (L), Cytotoxin (C) | Indication | Phase | Efficacy | Side Effect | Reference |
|---|---|---|---|---|---|---|
| MEDI2228 | A: BCMA<br><br>L: Cleavable linker<br><br>C: PBD dimer | Relapsed/Refractory Multiple Myeloma | I | *N/A (Ongoing)*<br>*Last Update Posted: 11 November 2021*<br>*Estimated Study Completion Date: 30 June 2022* | | NCT03489525<br><br>Status: Active, not recruiting<br><br>[99] |
| MEDI4276 | A: HER2; ERBB2<br><br>L: Maleimido-caproyl linker<br><br>C: MMETA (AZ13599185) | HER2-expressing Advanced Solid Tumours | I | <u>*N/A*</u><br>*Last Update Posted: 18 June 2019* | Nausea Fatigue Transaminitis | NCT02576548<br><br>Status: Completed<br><br>[100] |
| MEDI7247 | A: ASCT2<br><br>L: Cleavable linker<br><br>C: PBD dimer | Haematological Malignancies | I | Progression-Free Survival<br>AML pts: 0.8 months<br>DLBCL pts: 2.8 months<br><br>Overall Survival<br>AML pts: 1.3 months<br>DLBCL pts: 5.4 months<br><br>[101] | Thrombocytopenia Hypophosphataemia Anaemia Neutropenia<br><br><br><br>[101] | NCT03106428<br><br>Status: Completed<br><br>[101] |
| Mirvetuximab soravtansine (IMGN853) | A: FOLR1<br><br>L: Cleavable disulfide<br><br>C: DM4 | Ovarian, Endometrial, Non-small cell lung cancer | III | Cancer Antigen-125 Response<br>51%<br><br>Objective Response Rate<br>22% | Eye disorder GI disorder Neuropathy disorder | NCT02631876<br><br>Status: Completed<br><br>[102] |
| MORAb-202 | A: FOLR1<br><br>L: Cathepsin-cleavable linker<br><br>C: Eribulin mesylate | FRα- positive solid tumours | I | Objective Response Rate<br>Complete responses: 5%<br>Partial responses: 41%<br><br>Stable Disease<br>36%<br><br>[103] | Leukopenia Neutropenia Interstitial lung disease<br><br>[104] | NCT03386942<br><br>Status: Active, not recruiting<br><br>[103] |
| Naratuximab emtansine (IMGN529) | A: CD37<br><br>L: Non-reducible thioether bond<br><br>C: DM1 | Non-Hodgkin's Lymphoma and Chronic Lymphocytic Leukaemia | I | Overall Response Rate:<br>13%<br>Complete Responses: 1<br>Partial Responses: 4<br><br>[105] | Fatigue Neutropenia Pyrexia Thrombocytopenia Febrile neutropenia Peripheral neuropathy<br><br>[105] | NCT01534715<br><br>Status: Completed<br><br>[106] |

**Table 3.** *Cont.*

| Compound Name | Target Antigen (A), Linker (L), Cytotoxin (C) | Indication | Phase | Efficacy | Side Effect | Reference |
|---|---|---|---|---|---|---|
| OBI-999 (Anti-Globo H ADC) | A: Globo H<br><br>L: Site-specific ThioBridge<br><br>C: MMAE | Locally advanced solid tumours | I/II | *N/A (Ongoing)*<br>*Last Update Posted: 16 December 2021*<br>*Estimated Study Completion Date:*<br>*9 December 2023* | | NCT04084366<br><br>Status: Recruiting<br><br>[107] |
| Patritumab deruxtecan (U3-1402, HER3-DXd) | A: HER3<br><br>L: Cleavable peptide<br><br>C: Deruxtecan | EGFR inhibitor-resistant and EGFR-mutated non-small cell lung cancer | I | <u>Objective Response Rate</u><br>Confirmed: 39%<br>(2% complete responses, 37% partial responses)<br><br><u>Time to Response</u><br>2.6 months | Thrombocyt openia Neutropenia Fatigue | NCT03260491<br><br>Status: Active, not recruiting<br><br>[108] |
| Pinatuzumab Vedotin (DCDT2980S) | A: CD22<br><br>L: Cleavable mc–val–cit–PABC<br><br>C: MMAE | Diffuse Large B-Cell Lymphoma | II | <u>Overall Survival</u><br>16.493 months<br><br><u>Progression-Free Survival</u><br>5.388 months | Fatigue Peripheral neuropathy Diarrhoea Neutropenia | NCT01691898<br><br>Status: Completed<br><br>[109] |
| PSMA ADC<br><br>(Prostate-Specific Membrane Antigen Antibody Drug Conjugate) | A: PSMA<br><br>L: Cleavable Dipeptide<br><br>C: MMAE | Prostate Cancer | II | <u>PSA Response</u><br><u>(>50% Decrease)</u><br>Chemotherapy-experienced:11%<br><br>Chemotherapy-naive: 21%<br><br><u>Overall Radiologic</u><br><u>Response</u><br><u>(Stable Disease)</u><br>Chemotherapy-experienced: 61%<br>Chemotherapy-naive: 69% | Anaemia GI disorders Fatigue Decreased appetite. Peripheral neuropathy | NCT01695044<br><br>Status: Completed<br><br>[110] |
| Rovalpitu zumab tesirine (Rova-T) | A: DLL3<br><br>L: Cleavable dipeptide<br><br>C: PBD dimer | Small-cell lung cancer | III | <u>Overall Survival</u><br>Rova-T/Dexamethasone:<br>8.8 months<br><br>Placebo<br>9.89 months | Thrombocyt openia Pericardial effusion Abdominal pain Pneumonia | NCT03033511<br><br>Status: Terminated<br><br>[111] |
| SAR566658 | A: CA6<br><br>L: SPDB<br><br>C: DM4 | Neoplasm Malignant | I | <u>Stable Disease</u><br>39%<br><br>[112] | Fatigue Peripheral neuropathy GI disorders Neutropenia<br><br>[112] | NCT01156870<br><br>Status: Completed<br><br>[113] |

**Table 3.** *Cont.*

| Compound Name | Target Antigen (A), Linker (L), Cytotoxin (C) | Indication | Phase | Efficacy | Side Effect | Reference |
|---|---|---|---|---|---|---|
| | | Triple-Negative Breast Cancer | II | - | Keratitis Asthenia Keratopathy | NCT02984683<br><br>Status: Terminated (Limited clinical benefit combined serious ophthalmological event)<br><br>[114] |
| SKB264 | A: TROP2<br><br>L: *N/A*<br><br>C: Belotecan derivatives | Locally advanced unresectable/ metastatic solid tumours | I/II | *N/A (Ongoing)*<br>*Last Update Posted: 11 January 2022*<br>*Estimated Study Completion Date: December 2022* | | NCT04152499<br><br>Status: Recruiting<br><br>[115] |
| STRO-001 | A: CD74<br><br>L: Non-cleavable dibenzocyclooctyne (DBCO) linker<br><br>C: Maytansoid | B-Cell Malignancies | I | *N/A (Ongoing)*<br>*Last Update Posted: 19 October 2020*<br>*Estimated Study Completion Date: November 2023* | | NCT03424603<br><br>Status: Recruiting<br><br>[116] |
| SYD1875 | A: 5T4<br><br>L: Cleavable valine-citrulline-seco<br><br>C: Duocarmycin analogues | Solid tumour | I | *N/A (Ongoing)*<br>*Last Update Posted: 13 January 2022*<br>*Estimated Study Completion Date: January 2024* | | NCT04202705<br><br>Status: Active, not recruiting<br><br>[117] |
| Telisotuzumab vedotin<br><br>(ABBV-399) | A: ABT-700<br><br>L: Cleavable dipeptide<br><br>C: MMAE | Advanced solid tumours cancer and non-small cell lung cancer | I | *N/A (Ongoing)*<br>*Last Update Posted: 1 February 2022*<br>*Estimated Study Completion Date: 26 December 2024* | | NCT02099058<br><br>Status: Recruiting<br><br>[118] |
| Trastuzumab Deruxtecan<br>(DS-8201a) | A: ERBB2<br><br>L: Cleavable tetrapeptide linker<br><br>C: Topoisomerase I inhibitor | HER2-Low Advanced Breast Cancer | III | <u>Progression-Free Survival</u><br>9.9 months<br><br><u>Overall Survival</u><br>23.4 months | Nausea Alopecia Neutropenia Anaemia Fatigue | NCT03734029<br><br>Status: Active, not recruiting<br><br>[119] |

**Table 3.** *Cont.*

| Compound Name | Target Antigen (A), Linker (L), Cytotoxin (C) | Indication | Phase | Efficacy | Side Effect | Reference |
|---|---|---|---|---|---|---|
| Trastuzumab duocarmazine (SYD985) | A: HER2<br><br>L: Cleavable linker<br><br>C: Duocarmycin | Metastatic Breast Cancer | III | *N/A (Ongoing)*<br>*Last Update Posted: 12 January 2022*<br>*Estimated Study Completion Date: May 2022* | | NCT03262935<br><br>Status: Active, not recruiting<br><br>[120] |
| TRPH-222 | A: CD22<br><br>L: 4AP non-cleavable linker<br><br>C: Maytansine | Relapsed and/or Refractory B-Cell Lymphoma | I | *N/A (Ongoing)*<br>*Last Update Posted: 1 April 2021*<br>*Estimated Study Completion Date: 30 August 2022* | | NCT03682796<br><br>Status: Recruiting<br><br>[121] |
| Upifitamab rilsodotin (XMT-1536) | A: NaPi2b<br><br>L: Cleavable linker<br><br>C: Auristatin F | Ovarian cancer and non-small cell lung cancer | I | *N/A (Ongoing)*<br>*Last Update Posted: 24 September 2021*<br>*Estimated Study Completion Date: 31 December 2022* | | NCT03319628<br><br>Status: Recruiting<br><br>[122] |
| W0101 | A: IGF-1R<br><br>L: Non-cleavable maleimido-caproyl linker<br><br>C: Auristatin derivative | Advanced or metastatic solid tumours | I/II | *N/A (Ongoing)*<br>*Last Update Posted: 12 August 2021*<br>*Estimated Study Completion Date: 30 December 2024* | | NCT03316638<br><br>Status: Recruiting<br><br>[123] |
| Zilovertamab Vedotin (MK-2140) | A: ROR1<br><br>L: Cleavable mc–val-cit–PABC linker<br><br>C: MMAE | Relapsed or Refractory Diffuse Large B-Cell Lymphoma | II | *N/A (Ongoing)*<br>*Last Update Posted: 26 January 2023*<br>*Estimated Study Completion Date: 10 June 2025* | | NCT05144841<br><br>Status: Recruiting |

Abbreviation: MMAF, Monomethyl auristatin F; SPDB, N-succinimidyl-4-(2-pyridyldithio)butyrate; DM4, Ravtansine; pAcF, p-acetyl phenylalanine; MMAE, Monomethyl auristatin E; Ror2, Receptor tyrosine kinase-like orphan receptor 2; CD25, Cluster of Differentiation 25; PBD dimer, Pyrrolobenzodiazepine dimer; CD19, Cluster of Differentiation 19; CD71, Cluster of Differentiation 71; TACSTD, tumour-associated calcium signal transducer; EGFR, epidermal growth factor receptor; B7-H3, B7-Homolog 3; Cleavable mc–val–cit–PABC, Cleavable maleimidocaproxyl–valine–citrul–ine-p-aminocarbamate; gpNMB, Glycoprotein nonmetastatic melanoma protein B; CD79B, Cluster of Differentiation 79B; CD56, Cluster of Differentiation 56; DM1, Mertansine, N2′-deacetyl-N2′-(3-mercapto-1-oxopropyl)-maytansine; BCMA, B-cell maturation antigen; ASCT2, alanine–serine–cysteine transporter 2; FOLR1, Folate receptor 1; CD37, Cluster of Differentiation 37; Globo H, globohexaosylceramide; HER2, human epidermal growth factor receptor 2; HER3, human epidermal growth factor receptor 3; CD22, Cluster of Differentiation 22; PSMA, Prostate-specific membrane antigen; DLL3, delta-like ligand 3; CA6, Carbonic Anhydrase 6; TROP2, trophoblast antigen 2; CD74, Cluster of Differentiation 74; 4AP non-cleavable linker, 4-Aminopyridine non-cleavable linker; NaPi2b, Sodium-dependent phosphate transport protein 2b; IGF-1R, insulin-like growth factor 1 receptor; ENPP3, Ectonucleotide Pyrophosphatase/Phosphodiesterase 3; AST, aspartate aminotransferase; ALT, Alanine transaminase; GI disorder, Gastrointestinal disorder; pts, patients; AML, Acute Myeloid Leukemia; DLBCL, Diffuse Large B-cell Lymphoma; q3wk, once every three weeks; q6wk, once every six weeks; FLT3, FMS-like tyrosine kinase 3; ROR1, Receptor tyrosine kinase-like orphan receptor 1.

*4.1. Potential ADCs to Be Approved*

4.1.1. AGS67E

AGS67E is a humanized IgG2 designed to target CD37. It is an ADC linked to the cytotoxic MMAE via a protease cleavable linker. AGS67E is able to exert direct proapoptotic activity and has potent cytotoxicity towards several non-Hodgkin lymphomas (NHLs) and acute myeloid leukemia (AML).

A phase I clinical trial (NCT02175433) had been conducted to assess the safety, tolerability and pharmacokinetics of AGS67E in patients with relapsed or refractory lymphoid malignancies. A total of 71 participants were enrolled in this study. AGS67E had a moderate response in 50 patients with R/R B-NHL and R/R T-NHL. The overall response rate was 22%, and the complete response rate was 14%. The most reported adverse effects were peripheral neuropathy and neutropenia. The results indicated that AGS67E showed positive effects in patients with relapsed or refractory lymphoid malignancies. Thus, further investigation of AGS67E can be continued [70].

4.1.2. Denintuzumab Mafodotin (SGN-CD19A)

Denintuzumab mafodotin (SGN-CD19A) is an ADC designed to target CD19. It is a humanised monoclonal antibody conjugated to the cytotoxic monomethyl auristatin F (MMAF) via a non-cleavable linker. Upon internalisation, MMAF will be released into the cells, which bind to the tubulin and inhibit its polymerisation. Eventually, the apoptosis of the tumour cell will occur and thus inhibit the cell growth of CD19-expressing malignancies [124].

A phase I clinical trial (NCT01786135) had been completed to assess the safety and tolerability of SGN-CD19A among 64 patients with relapsed or refractory B-lineage non-Hodgkin lymphoma (B-NHL) [83]. A total of fifty-two patients were treated every three weeks (0.5–6 mg/kg), and ten patients were treated in the q6wk schedule (3 mg/kg). In the dosing schedule, of 22 relapsed patients, seven patients had complete remission while four patients had partial remission (the complete remission rate and the objective response rate were 32 and 50%, respectively). In addition, of the 29 refractory patients, seven patients had stable diseases (objective response rate and complete remission rate was 21 and 10%, respectively). However, the most frequently reported adverse effects were blurry vision, dry eye, fatigue, keratopathy, constipation, photophobia, and nausea. Nevertheless, keratopathy could be managed with topical steroids and dose modifications. The conditions would be improved and resolved within five weeks [125].

Another phase I clinical trial (NCT01786096) had also been performed to evaluate the safety and tolerability of SGN-CD19A among 92 patients with relapsed or refractory B-lineage acute lymphoblastic leukaemia (B-ALL), B-lineage lymphoblastic lymphoma (B-LBL), Burkitt lymphoma, or leukaemia [84]. There were 71 patients were treated, of which 59 suffered from relapsed or refractory B-lineage acute lymphoblastic leukaemia. Of 59 patients, 32 were treated weekly (0.3–3 mg/kg), while 23 were treated in the q3wk schedule (4–6 mg/kg). Six patients who received the treatment once a week had complete remission; one had partial remission (the complete remission rate was 19%). In contrast, eight patients who received the treatment once every three weeks had complete remission, but three had incomplete platelet recovery, and two had incomplete blood recovery. These results indicated that the complete remission rate was 35%, higher than the patients treated weekly. The most reported adverse events were pyrexia, nausea, fatigue, headache, chills, vomiting, blurred vision, and anaemia. The results demonstrated that the denintuzumab mafodotin expressed promising activity in patients with B-ALL, B-LBL, Burkitt lymphoma or leukaemia. Hence, future clinical investigation and development of SGN-CD19A can be pursued [125].

4.1.3. MEDI4276

MEDI4276 is an ADC consisting of a HER2-bispecific antibody that targets HER2-positive tumours. It is conjugated to AZ13599185, a tubulysin-based microtubule inhibitor

via a maleimidocaproyl linker. The toxin AZ13599185 inhibits microtubule polymerization during mitosis and eventually induces cell death. MEDI4276 has potent antitumour activity in HER2-low cell lines that are refractory to Trastuzumab emtansine (T–DM1) treatment [126]. A phase I/II clinical trial (NCT02576548) had been performed to evaluate the safety, pharmacokinetics, immunogenicity, and antitumour activity of MEDI4276 in patients with select HER2-expressing advanced solid tumours. A total of 43 patients with advanced, HER2-overexpressing breast and gastric cancers received escalating doses every three weeks (0.05–0.9 mg/kg). However, 38 patients reported having adverse events, the most commonly reported adverse events including nausea, fatigue and transaminitis. The maximum tolerated dose (MTD) of MEDI4276 was recorded at 0.9 mg/kg. In view of the severe toxicities, the clinical testing of this agent has been discontinued [100].

### 4.1.4. Mirvetuximab Soravtansine (IMGN853)

Mirvetuximab soravtansine (IMGN853) is an immunoconjugate consisting of M9346A (humanised monoclonal antibody), linked by the cleavable disulfide to the cytotoxic DM4. The monoclonal antibody moiety of IMGN853 recognises and binds to antigen folate receptor 1 (FOLR1, folate receptor alpha). Upon the antibody–antigen interaction and internalisation, IMGN853 releases its cytotoxic payload (DM4), which binds to tubulin and disrupts the assembly of microtubules [127].

A phase III clinical trial (NCT02631876) had been conducted to access the efficacy and safety profile of mirvetuximab soravtansine versus the selected single-agent chemotherapy (Paclitaxel, Pegylated liposomal doxorubicin, Topotecan) in women with primary peritoneal cancer and/or fallopian tube cancer and platinum-resistant folate receptor alpha positive advanced epithelial ovarian cancer [128]. A total of 366 patients were enrolled in this study: 248 patients received mirvetuximab soravtansine, whereas 118 patients received single-agent chemotherapy as treatment. The study result showed that mirvetuximab soravtansine possessed better efficacy than the selected single-agent chemotherapy. The objective response rate in the mirvetuximab soravtansine-treated group was 22%, which was 10% higher than in patients receiving single-agent chemotherapy. Additionally, the cancer antigen response rate in patients receiving mirvetuximab soravtansine was 51%, whereas in patients receiving single-agent chemotherapy was only 27%. However, reported blurred vision and keratopathy can be resolved with prophylactic corticosteroid eye drops [129]. Given the positive results, the FDA will likely approve this new agent shortly.

### 4.1.5. Patritumab Deruxtecan (U3-1402)

Patritumab deruxtecan (U3-1402 or HER3-DXd) is an ADC consisting of the fully humanised monoclonal antibody patritumab (known as AMG 888 or U3-1287) linked to the cytotoxic payload deruxtecan (topoisomerase I inhibitor) via a cleavable peptide linker. Patritumab specifically targets HER3 which is overexpressed on certain cancerous cells. When monoclonal antibody moiety binds to HER3 on the cell membrane, internalisation will occur, and the linker will be cleaved, releasing the payload deruxtecan. Deruxtecan induces tumour cell death through DNA damage [130].

A phase I clinical trial (NCT03260491) was conducted to evaluate the safety and antitumour activity of patritumab deruxtecan in patients with non-small cell lung cancer. This study consists of two parts, dose escalation and dose expansion. A total of 216 participants were enrolled and allocated into three cohorts with different doses to determine the recommended dose. The intravenous dose of 5.6 mg/kg every three weeks demonstrated great antitumour activity in metastatic or locally advanced EGFR mutation NSCLC [131]. At data cut-off, 57 patients of 216 treated with patritumab deruxtecan 5.6 mg/kg IV every three weeks demonstrated a median follow-up of 10.2 months. The confirmed objective response rate was 39% (22 out of 57), with 14 patients' responses occurring within three months of treatment. The disease control rate was 72%, with a median progression-free survival was 8.2 months. A phase II clinical trial (NCT04619004) is currently being initiated to study the efficacy of patritumab deruxtecan in patients with EGFR-mutated NSCLC after

the failure of EGFR tyrosine kinase inhibitor therapy and platinum-based chemotherapy. The study is estimated to be completed by 2024 [132].

### 4.1.6. Telisotuzumab Vedotin (ABBV-399)

Telisotuzumab vedotin (ABBV-399) is an ADC containing monoclonal antibody ABT-700, linked to cytotoxin MMAE (monomethyl auristatin E) via cleavable dipeptide. ABT-700 targets the c-Met receptor, a tyrosine kinase receptor that is overexpressed in patients with NSMCLC (non-small cell lung cancer). Preclinical data demonstrate that ABBV-399 can directly deliver cytotoxin to tumour cells by overexpressing the c-Met receptor [133]. A phase I clinical trial (NCT02099058) had been conducted to evaluate its pharmacokinetics, safety profile and preliminary efficacy as monotherapy compared to in combination with Erlotinib, Osimertinib, and Nivolumab in patients likely to express c-Met receptor the advanced solid tumours cell surface. A total of 225 participants are enrolled, and the study has been estimated to be completed by 2025 [118].

### *4.2. Discontinued/Terminated Clinical Trials*
### 4.2.1. AGS-16C3F

AGS-16C3F is a novel ADC designed to target ENPP3, also known as cell-surface ectonucleotide pyrophosphatase/phosphodiesterase 3 [134]. AGS-16C3F is a fully humanised monoclonal antibody conjugated to MMAF (Monomethyl auristatin F) through a non-cleavable maleimido–caproyl linker. MMAF is an auristatin derivative, a potent microtubule disruptive agent with antineoplastic activity. Upon binding to ENPP3, internalisation occurs, followed by proteolytic cleavage of the linker to release MMAF in the cells. It will induce tumour cell apoptosis by inhibiting tubulin polymerisation and lead to G2/M phase arrest.

A phase II clinical trial (clinical trials identifier NCT02639182) has been conducted to assess the PFS of AGS-16C3F and axitinib in patients with metastatic renal cell carcinoma. A total of 133 participants were registered in this study [68]. Eighty-four patients had completed the trial at the data cut-off in the total population. The overall survival with AGS-16C3F was 13.1 months, and the progression-free survival was 2.9 months. However, some common adverse effects have been reported, including fatigue, nausea, and blurred vision. On the other hand, patients with metastatic renal cell carcinoma that received axitinib presented more tolerable side effects such as fatigue and diarrhoea. Since AGS-16C3F did not meet the primary endpoint of this study, the future development of AGS-16C3F in patients with metastatic renal cell carcinoma would not be continued [134].

### 4.2.2. Coltuximab Ravtansine (SAR3419)

Coltuximab ravtansine, also known as SAR3419, is an ADC that targets CD19 (B-lymphocyte antigen CD19). SAR3419 is a humanised monoclonal antibody conjugated to cytotoxic DM4 (maytansinoid) through a cleavable disulfide bond. CD19 is a type 1 transmembrane glycoprotein ubiquitously expressed in B cell malignancies. Internalisation will occur upon binding to the CD19 antigen on the target cells, releasing DM4 into the cells. This will lead to microtubule disruption and cell cycle arrest and hence eventually causes cell death [135].

A phase II clinical trial (clinical trials identifier NCT01472887) has been conducted to investigate the safety and efficacy of SAR3419 in patients with diffuse large B-cell lymphoma. A total of 61 participants were recruited in this study [77]. Forty-one patients received coltuximab ravtansine, and the overall response rate was 43.9%, while overall survival, progression-free survival and median duration of response were 9.2 months, 4.4 months and 4.7 months, respectively. Nevertheless, some reported adverse effects included asthenia and fatigue, nausea, and diarrhoea. As a result, coltuximab ravtansine was well tolerated and showed moderate clinical responses in diffuse large B-cell lymphoma [136]. Likewise, another phase II clinical trial (NCT01440179) has also been conducted to test the safety and efficacy of SAR3419 in patients with acute lymphocytic leukaemia [80]. A total

of 36 patients were recruited, of which 19 patients were treated during dose escalation, whereas 17 patients were treated during dose expansion. Four out of 17 patients achieved a 25.47% objective response rate for the primary endpoint. Moreover, for the secondary endpoint, the duration of response was short, which was 1.94 months. The most frequent adverse effects were pyrexia, diarrhoea, nausea, and hematologic disorders such as thrombocytopenia, anaemia, and lymphopenia. The research had been terminated due to the limited efficacy in patients with acute lymphocytic leukaemia [135].

### 4.2.3. Glembatumumab Vedotin (CDX-011)

Glembatumumab vedotin (CDX-011 or CR011-vcMMAE), an ADC designed to target glycoprotein NMB (gpNMB) that is often overexpressed in human malignant tissues, including melanoma and breast cancer. The high expression of gpNMB is associated with invasion and metastasis. CDX-011 links a CR-011, the gpNMB targeting antibody, to the MMAE (monomethyl auristatin E) through a cleavable dipeptide linker. Upon binding and internalisation of CDX-011, the release of MMAE induces cell death by disrupting the microtubules [137].

Several clinical trials have been conducted to evaluate the safety and efficacy of CDX-011. A phase II clinical trial (NCT02302339) had been performed to assess its safety and effectiveness as monotherapy and in combination with immunotherapies in 132 patients with advanced melanoma [93]. A sum of 62 patients received CDX-011. The objective response rate, response duration, and overall survival were 11%, 6 months, and 9 months, respectively. The most commonly reported adverse events were alopecia, neuropathy, rash, fatigue, nausea, and neutropenia. In this trial, glembatumumab vedotin was considerably tolerated but displayed a modest efficacy against the tumour cell [137].

### 4.2.4. Lorvotuzumab Mertansine (IMGN-901)

Lorvotuzumab mertansine, also known as IMGN-901, is an ADC used to target CD56 (also known as neural cell adhesion molecule 1 or NCAM-1). IMGN-901 is a humanised monoclonal antibody conjugated to cytotoxic DM1 (mertansine, maytansinoid) via a disulfide linker. CD56 is a membrane glycoprotein expressed in malignant tissues, such as small cell lung cancer (SCLC) and leukaemia. Upon binding to the CD56 on the tumour cell surface, IMGN-901 will be internalised and cleaved to release DM1 into the cells, which causes tumour cell death by inhibiting tubulin polymerisation [138].

A phase I/II clinical trial (NCT01237678) had been conducted to assess the safety and efficacy of IMGN-901 in combination with carboplatin and etoposide in patients with extensive-stage small-cell lung cancer (SCLC) [98]. A sum of 181 participants was enrolled in this study, of which 33 participants were registered in the phase I trial while 148 participants were recruited in the phase II trial. In the phase II trial, the combination regimen's overall survival and progression-free survivals were 10.1 months and 6.2 months, respectively. In contrast, for carboplatin and etoposide alone, the overall and progression-free survivals were 11 months and 6.7 months, respectively. Additionally, peripheral sensory neuropathy was frequently reported as a side effect among patients who received the combination regimen, which accounted for 61%, and the most observed side effects were anaemia, neutropenia and fatigue. The study had been discontinued due to the reported toxicities and limited efficacy of the combined regimen [138].

### 4.2.5. Rovalpituzumab Tesirine (Rova-T)

Rovalpituzumab Tesirine, known as Rova-T, is an ADC containing monoclonal antibody (humanised IgG1) conjugated to the cytotoxic PBD dimer via cleavable dipeptide. The antibody moiety of Rova-T selectively recognises and binds to the delta-like protein (DLL3) on the tumour cell's surface. The dipeptide linker is cleaved upon internalisation to release the cytotoxin. Cytotoxin PBD dimer binds and induces DNA strand breakage [139]. A phase III clinical trial (NCT03033511) had been conducted to evaluate the efficacy of rovalpituzumab tesirine as maintenance therapy following first-line platinum-based chemother-

apy. The result demonstrated no survival benefit of rovalpituzumab tesirine compared to placebo in treating small cell lung cancer. The overall survival in patients receiving a placebo was 9.79 months, higher than those receiving Rova-T, 8.48 months. Due to this unfavourable consequence, AbbVie officially announced discontinuing theRova-T research and development program [140].

## 5. Challenges of the Use of Antibody–Drug Conjugate

The development of a new drug is never an easy journey. The early stage ADC development faced a lot of barriers such as unstable linkages, the low toxicity level of payload, short blood residency time, off-target drug toxicity, high immunogenicity, and low level of penetration into the tumour microenvironment (Tables 4 and 5) [141]. Through these years, after tons of studies and funds contributed to ADC development, several ADCs have been approved for clinical use.

**Table 4.** Summary of ADC challenges.

| Main Topics | Summary Points |
|---|---|
| Toxicity of ADC | • Ultra-toxicity due to the subnanomolar range of payload<br>• Payloads have bystander killing effects on both cancer and non-cancerous cells<br>• The toxicity of ADC can be improved by maximising the therapeutic index and adjusting the ADC dosage<br>• The drug-antibody ratio (DAR) should be optimised in a suitable range for each ADC activity |
| Antibody & Antigen Specificity | • Select antigen that is highly expressed in tumours<br>• Homogenous antigen is of an advantage over heterogeneous antigen<br>• There may be a difference in antigen structure between human and selected animal |
| Immunogenicity of Antibody | • Murine antibody can trigger immunogenic response (First generation ADC)<br>• Humanised monoclonal antibodies (mouse chimeric antibody) can reduce the immunogenic concern (Second generation ADC)<br>• Antibody with long-half life and highly stable is desired |
| Stability of linkers | • Cleavable linkers can be affected by the physiological system<br>• Non-cleavable linkers are more stable and perform better than cleavable linkers<br>• The stability can be improved with integrated linker design software. |

### 5.1. Toxicity of ADC

ADCs were initially designed to reduce systemic toxicities in comparison to conventional chemotherapy by targeting antibodies on highly expressed cancer biomarkers. However, toxicity remains one of the most intimidating challenges faced during the use of ADCs [27]. Chemotherapeutic drugs often cause toxicities even in the normal range of therapeutic doses [142], more so for ADCs that carry a subnanomolar range of payloads. It should be noted that ADCs cannot carry a large amount of cytotoxic payload due to the conformation and structure of the payload and antibodies. If a less potent payload is used in the development of an ADC, a higher dose is required to achieve the therapeutic effect, which may increase the risk of toxicity of the treatment. Thus, a more potent and highly super toxic agent as the cytotoxic payload is ideal for ADC designation [141]. According

to [27], the issue of ADC toxicity could be improved by maximising the therapeutic index and adjusting the ADC dosage. The therapeutic index could be maximised in several ways, including integrating maytansine derivatives into ADCs, or biomarkers could be implemented to choose the correct patient population, observe initial response signals, or guide the combination therapy. Continuous monitoring should be carried out to observe the patient's response.

**Table 5.** Advantages and Disadvantages of ADC.

| Advantages | Disadvantages |
|---|---|
| ADC has increased the tumour specificity and selectivity and enhances the delivery of the cytotoxic payload to the cancer cell. | ADC is unable to carry a large amount of cytotoxic payload which in turn required a higher dose to achieve the therapeutic effect leading to an increased risk of toxicity. |
| ADC can carry a very toxic payload that was previously discarded due to safety issues. | ADC bystander killing effect. |
| With a stable linker, the therapeutic window of ADC will be expanded even if higher dosages of toxin-carrying ADC be administered to patients. | The nonspecific binding of inappropriate selection of antibodies might lead to undesirable toxicity. |

The DAR is a crucial therapeutic index component for ADR. The DAR of ADCs approved up to this point ranged between 2 to 8, with four drug molecules/ADC (DAR = 4) being the most common DAR [143]. From a technical point of view, a high DAR value above 8 has a detrimental effect on ADCs' efficacy, causing premature destabilisation, aggregation, increased off-target toxicity, and enhanced drug clearance from systemic circulation [144]. While drugs with low DAR value were expectedly less potent than high DAR, causing an impact in terms of the therapeutic efficacy of ADCs. Thus, the DAR should be optimised in a suitable range for each ADC activity.

Another concern is bystander killing activity, which occurs when the ADC drug is released into the extracellular environment or when the ADC drug is released from the target cell after internalisation and hydrolysis. Both circumstances result in the drug being taken up and killing surrounding bystander cells, which may or may not express the ADC target antigen [145]. There is an effort to reduce the bystander killing effects of ADCs. However, this ADC is usually less potent or leads to short-lasting responses [146]. On the other hand, they are generally more well-tolerated. The auristatins MMAE and MMAF are two payloads that have been thoroughly studied for their differing characteristics and capacities to induce bystander killing [146]. Due to its cell permeability and lipophilicity, MMAE has been found to cause bystander killing of neighbouring antigen-negative cells. In contrast, MMAF has not been demonstrated to cause bystander killing, which has been explained by its reduced cell permeability coupled with its more excellent hydrophilicity [146]. Based on the findings, research is ongoing to increase the high drug loading on antibodies with improved efficacy and reduced bystander effects. For instance, a novel dolaflexin ADC has high DAR and potency with controlled bystander activity is developed. It is hoped that this breakthrough could eradicate the widely reported dose-limiting toxicity of neutropenia associated with the existing bystander impacts, particularly on auristatin ADC platforms.

*5.2. Antibody & Antigen Specificity*

Ideally, the antibody of an ADC should be highly specific and bind to the targeted antigen only. The purpose of having a highly specific antibody is to avoid any cross-reaction by binding to other antigens that might lead to undesirable toxicity [141]. Indeed, the targeted antigen should be highly expressed only in tumours but absent or minimally expressed in non-cancerous cells. This makes the targeted antigen tumour-specific, and the therapeutic effect can be specifically carried out on tumour cells without harming other non-cancerous cells. Sadly, most of the discovered antigens are also expressed in both cancerous and non-cancerous cells; the only difference is the level is higher in the

cancerous cells than in non-cancerous cells [27]. For instance, the SGN-15, also termed BR–96 doxorubicin, leads to haemorrhagic gastritis because the targeted antigen, Lewis Y, is also expressed in the gastric mucosa. Despite this discouraging statement, some of the antigens that have been used in the development of ADCs include folate receptor alpha (FRα), mesothelin, MUC-16 (CA-125), NaPi2b, and human tissue factor. Another concern is that there may be a difference in antigens between humans and rodents. It is suggested to select the animal model for preclinical studies with an antigen similar to humans [27].

The homogeneity or heterogeneity expression of antigen on the cell surface also determines the ADC activity. [147] stated that a homogenous antigen will conjugate at the same position in every mAb molecule, while a heterogeneous antigen will conjugate at a random position which differs in every mAb molecule. The homogenously expressed antigen is more desired in ADCs due to enhanced ADC targeting, batch-to-batch consistency, and higher desirability from a regulatory perspective [147]. A heterogeneous antigen isn't desired in ADC; it could increase bystander killing activity. The bystander killing activity is beneficial to increase the killing rate of cancer cells but gives a poorer safety profile as the non-cancerous cells may be harmed simultaneously. Site-specific conjugation methods are developing to produce homogenous ADCs targeting just the homogenous antigen on cancer cells [147].

Since the tumour microenvironment is critical for tumour growth, the tumour stromal compartment is becoming a growing area of focus for ADC targeting development [148]. As tumour-associated stroma from various cancer types shares similar traits and characteristics, it gives the possibility to broaden ADC therapeutic utility beyond solely tumour-targeting approaches, which are limited to groups of antigen-positive patients only [10]. A recent finding discovered that ADC efficacy was bolstered by an unexpected cell death mechanism known as DAaRTS, in which TEM8–ADC was internalised by drug-resistant TEM8+ tumour stromal cells to release cell-permeable MMAE-free drug to cause rapid bystander killing of nearby drug-sensitive tumour cells in a target-independent manner [148]. Fibroblast activation protein (FAP), which was initially thought to be a stromal cell target, was expressed in healthy tissue, limiting the anticipation for FAP targeting. Fortunately, breakthroughs of barriers revealed antigens ubiquitously expressed on tumour-associated stroma and may possess the specificity required to build a stromal cell-directed ADC [148]. Even though the potential of stromal targeting has long been known, potential vulnerabilities provided by stromal cells have yet to be exploited, owing to a scarcity of appropriate targets with high tumour selectivity [10].

*5.3. Immunogenicity of Antibody*

Antibodies with high immunogenicity will provoke an immune response when a therapeutic agent is administered. It was desired to have an antibody with minimal immunogenicity to avoid an immune response from the body. This issue could be improved by replacing murine antibodies with humanised monoclonal antibodies to minimise the risk of immunogenicity [149]. The capability to induce receptor-mediated internalisation was also an essential parameter of the desired antibodies. The rapid internalisation of antibodies could improve ADC's safety profile and therapeutic efficacy because it significantly decreases the possibilities of off-target release [141]. A highly stable antibody in blood circulation with a long half-life and a high molecular weight is desirable in ADC drugs. Thus, antibodies of ADC should have low immunogenicity and increased stability with rapid internalisation.

First-generation ADCs utilised murine antibodies against the target antigen. Still, the murine antibody elicited a significant immune response, causing many patients to develop anti-human antibodies, diminishing the medication's efficacy. Thanks to advances in gene engineering technology, this problem was rectified, resulting in second-generation ADCs. A murine antibody was turned into a mouse or humanised chimeric antibody [150]. The mouse light and heavy chain variable domains are linked to human constant regions in the chimeric antibody, which helps to minimise immunogenicity and human anti-mice

antibody. The therapeutic efficacy of second-generation ADCs seemed promising. For instance, the newly developed ADC comprising chimeric anti-CD30 mAb is connected to lidamycin via a non-protease peptide linker. This ADC exhibits a high efficacy, cytotoxicity and specific affinity against CD30-overexpressing tumour cells [150].

*5.4. Stability of Linkers*

Linkers are short bridges that bind the drug to the antibody covalently. The stability of linkers is a significant challenge faced in ADC use. In blood circulation, a stable linker will keep the drug tightly intact to the antibody and perform intercellular cleavage after reaching the targeted cell. Linker acts as a critical parameter in ADC designation as it influences the safety and efficacy of the drug in patients. An unstable linkage will degrade and release the drug before reaching the targeted cell, leading to off-target cytotoxicity. Thus, an ADC's safety and efficacy will significantly reduce with an unstable linkage. The two main types of linkers include cleavable and non-cleavable linkers. Although non-cleavable linkage performs better than the cleavable, the most suitable linkage for each ADC will be selected mainly based on its stability with the functional group of ADC and environment [149].

To exert their cytotoxic effects, Ado-trastuzumab emtansine, an ADC drug with a non-cleavable linker such as thioether, must be internalised by the target cell for the antibody and degraded before drug release. But since this type of ADC often comprises a potent tubulin-binding maytansine derivative which is DM1, the cleaved drug product would not reach neighbouring cells to exhibit bystander killing activity due to the low drug penetration (positively charged drug). As a result, non-cleavable ADCs are indeed efficacious against target antigen-expressing cells after internalisation and are favourable to malignancies with high and homogeneous antigen expression [145]. Despite the unpredictable translational consequences from pre-clinical to clinical trials, obtaining the most optimal dose would lead to a more precise therapeutic index assumption. It is recommended to use integrated linker design approaches to attain the highest intratumoural payload PK profiles by driving above the threshold dose [151].

*5.5. Target Features of Successful ADCs*

As shown in Table 4, the target features of successful ADCs can be identified from several aspects. First, the toxicity of a successful ADC must be acceptable, with lower bystander killing effects to protect normal cells from being destroyed unintentionally, and the DAR should be optimized for each ADC. Second, the interaction between antigen expressed in tumours and the antibody must be highly specified to achieve a favourable on-target effect. Moreover, since ADCs consist of engineered antibodies, they may trigger an immunogenic response. Thus, a successful ADC must be safe to introduce into the human body. Finally, the stability of the linker is one of the key features of successful ADC which should only release the drug when it interacts with antigen to promise the safety and efficacy of ADC. To summarize, a successful ADC should be safe, with acceptable toxicity, low or even non-immunogenicity, and the binding with the tumours antigen should be highly specified, and with a stable linker.

## 6. Computational Strategies to Address ADC's Challenges

The development of ADCs can be efficiently optimised by the application of computational approaches at each stage. For the development of specific antibodies with high affinities, understanding the complexes' conformational dynamics and three-dimensional (3D) structure is essential. Additionally, they enable a better comprehension of how antibodies interact with their antigens and ways to manipulate the interaction. To determine the ideal antigen target in silico, ML algorithms, molecular docking, and MD simulations can all be applied. More reliable quantum mechanics methodologies should be taken into consideration to clarify the molecular mechanism by which antibodies, cytotoxic medication, and target antigens interact with one another [152].

Through computational approaches, the biological and ADMET risk prediction for cytotoxic payload may be estimated. Molecular Dynamics (MD) simulations of antigens and antibodies, possible binding pockets for antibodies, and docking for the optimal structures of drugs, linkers, and ADCs are sought utilizing different computational techniques [153].

## 7. Neoantigens Cancer Vaccines

In addition to ADCs, neoantigen-targeted cancer vaccines had been seen as a great breakthrough in overcoming cancer. It utilizes the nature of the human immune system as recognizing and targeting tumour molecules, known as neoantigens, to kill a cancer cell. As compared to ADCs, neoantigen-targeted cancer vaccines are said to give active immunotherapy. It eliminates cancer cells by stimulating the body's immune system rather than passive acceptance by an organism of antibodies, cytokines, or transformed immune cells that can directly act on the tumour [154].

### 7.1. Neoantigen

Neoantigens are antigens that are expressed in cancer cells, indicating the mutations that have occurred during the transformation process. Differing from tumour-associated antigens (TAA), neoantigens molecules are genuinely unique to the cancer cell and may be recognized by the immune system, which in turn leads to specific anti-tumour activity by the host [155].

Neoantigens can be classified into two categories, namely shared neoantigens and personalized neoantigens. As the name suggests, shared neoantigens are mutated antigens that are common across different cancer patients and not present in the normal genome. They are highly immunogenic and possess the potential to be screened for use as broad-spectrum therapeutic cancer vaccines for patients with the same mutated gene. Personalized neoantigens are mutated antigens that are specific to most neoantigens and completely different from individuals. Thus, the personalized neoantigen vaccine can only be specifically targeted to each patient [154].

### 7.2. Principle of Neoantigen Vaccines

Unlike common prophylactic vaccines, neoantigen-targeted vaccines are administered to cancer patients, which may be supplemented by appropriate adjuvants, to activate the patient's autoimmune response and kill the tumour cells [156]. Being non-autoantigens, neoantigens are exposed to Major Histocompatibility Complex (MHC) molecules, which subsequently trigger the body's anti-tumour immune response.

When a vaccine is administered, the neoantigens will be recognized by antigen-presenting cells (APCs), mainly dendritic cells (DCs). It enters DCs via endocytosis and is degraded and processed into fragmental peptides. The degraded polypeptide fragment is then transported to the endoplasmic reticulum (ER) and binds with MHC classI and classII molecules to form a surface peptide–MHC molecular complex (pMHC), which will be further transferred to the APCs cell membrane. The interaction between the pMHC complex and T-cell receptor (TCR) on T cells causes the antigenic peptide of the MHC molecule to be recognized and promotes the proliferation and differentiation of immune cells. During T cell development, CD4+ T will produce cytokines, which promote the maturation of CD8+ T cells into cytotoxic T lymphocytes (CTL), and induce apoptosis of tumour cells through the Perforin/Granulase pathway or Death receptor pathway. The apoptotic tumour cells release more tumour antigens, further stimulating the immune system of the body and prompting the immune system to kill tumour cells specifically [157].

### 7.3. Types of Neoantigen Vaccines

Currently, the neoantigens vaccines under development include those based on tumour cell lysates, nucleic acids, and peptides, which contain or can encode neoantigens [158]. In addition to being used independently as vaccines, these neoantigens formulations can be loaded onto DCs, forming DC-based neoantigen vaccines, which gives substantial benefits

over the unloaded vaccines. Although DNA can be easily manipulated by molecular engineering, it tends to rely more on electroporation, and it is also limited by its potential to integrate into the genome. As for RNA vaccines, the integrity of nucleic acid will be affected by RNase degradation, which further compromises the effectiveness of vaccines. In addition, synthetic long peptide (SLP) is easy to store and has low toxicity; however, appropriate adjuvants are required [159]. The mechanism of action of common neoantigen vaccines is illustrated in Figure 5.

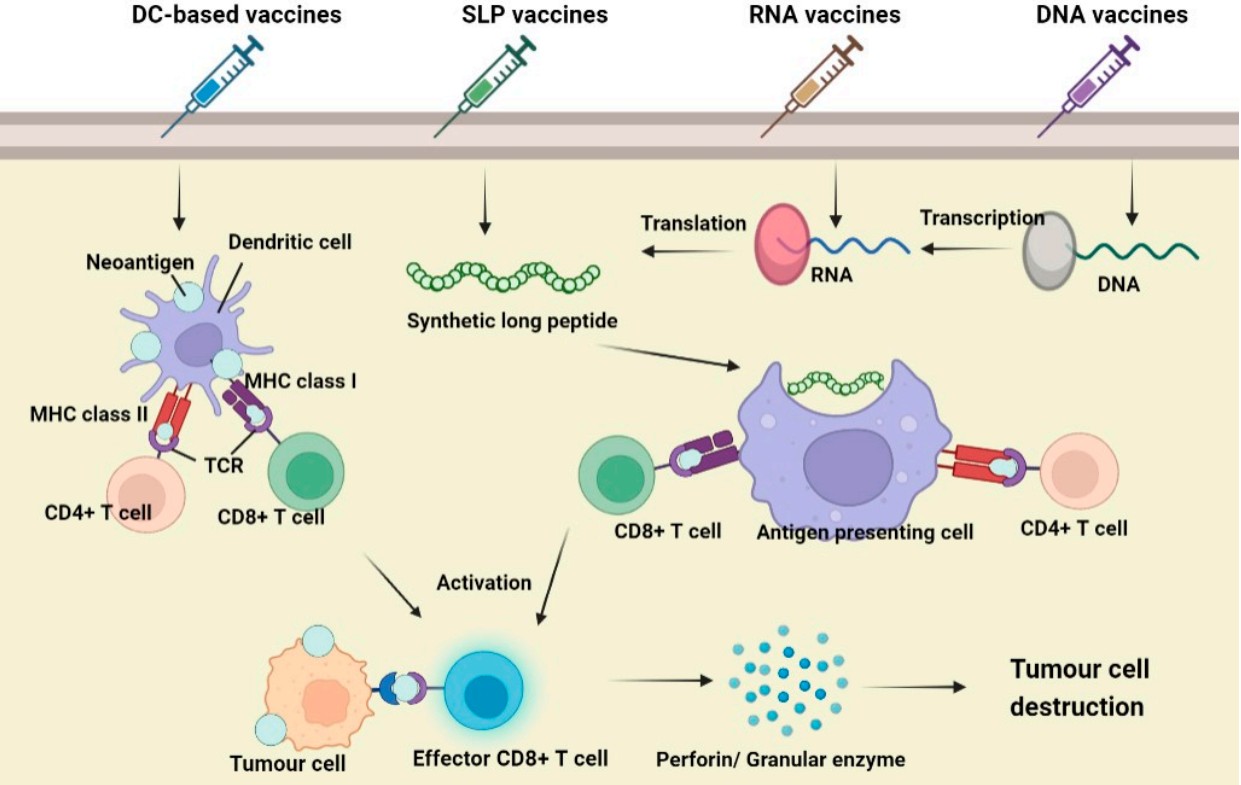

**Figure 5.** Mechanism of action of DC-based, SLP, RNA and DNA neoantigen vaccines (Created with https://www.biorender.com/) (accessed on 2 December 2022).

## 8. Conclusions

Antibody–Drug Conjugate is an advanced chemotherapy that has been developed for the treatment of relapsed and/or refractory cancer. High specificity, potency and efficacy are the main features of this antibody-based therapy. Technological advancements enable the optimum design of ADC with fully humanised or chimeric antibodies, selecting clinically functional linkers and using a subnanomolar range of compounds to treat cancer patients. With the accelerated approval from FDAs, many ADCs are now benefiting cancer patients in a hospital setting. Nevertheless, there is still room for improvement for ADCs, including ADC toxicity, specificity of antibodies and antigens, the immunogenicity of antibodies, and stability of linkers. Regardless of the challenges faced by ADC use, more new ADC interventions are being developed recently to improve ADC designation and solve these matters. With profuse research and investigation, promising ADCs will be produced, bringing aspiration to cancer patients.

**Author Contributions:** Conceptualization, J.B.F.; Data search and writing—original draft, Y.J.L., P.S.C.L., S.X.L., S.L.N., M.Y.O., H.M.P. and Z.Y.L.; writing—review and editing, H.Y.Y., S.B.H., R.S. and J.B.F. All authors have read and agreed to the published version of the manuscript.

**Funding:** This publication paper is funded by the Ministry of Higher Education (MOE) under Fundamental Research Grant Project (FRGS/1/2019/SKK08/TAYLOR/02/1). The figures in the article were generated from a BioRender account subscribed to by Taylor's University.

**Institutional Review Board Statement:** Not applicable.

**Informed Consent Statement:** Not applicable.

**Data Availability Statement:** Data are contained within the articles.

**Acknowledgments:** The authors would like to thank the laboratory staffs of Faculty of Health & Medical Sciences, Taylor University for their resources and technical support on the present study.

**Conflicts of Interest:** The authors declare no conflict of interest.

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
