# Peer review of "How Far Have We Developed Antibody–Drug Conjugate for the Treatment of Cancer?"

_ddc, doi:10.3390/ddc2020020_

Round 1
Reviewer 1 Report
The review does not report substantial novelty regarding other reviews published recently that, of course, are not included in the reference list such as:
"An Insight into FDA Approved Antibody-Drug Conjugates for Cancer Therapy" 10.3390/molecules26195847
"Antibody drug conjugate: the “biological missile” for targeted cancer therapy" 10.1038/s41392-022-00947-7
“Antibody–Drug Conjugates: The Last Decade” 10.3390/ph13090245
In addition, I would suggest the authors to be more careful drawing the molecular structures they did not follow a fixed style and everyone has a different size.
Reviewer 2 Report
This paper reviewed the principles of antibody-drug conjugates (ADCs) and the ADCs in clinic, under development and that discontinued/terminated in clinical trials. Additionally, the challenges about the use of ADC were discussed, highlighting the further directions for ADCs improvement. The topic fits the scope of the journal and may benefit the development of ADCs for the treatment of human diseases. In general, the manuscript is well-organized and the references can support the conclusions. The key issues are required to be addressed before its publication on Drugs and Drug Candidates.
1. The timeline of ADCs development is suggested to be introduced with a figure illustration in the Introduction section.
2. The target features of successful ADCs are suggested to be summarized in the section 5.
3. The advantages/disadvantages of ADCs with small-molecule drugs are suggested to be compared and analyzed (probably with a table).
Reviewer 3 Report
Antibody-drug conjugate for targeted cancer therapy is a promising area of research. I appreciate the authors for their efforts to assemble up-to-date information in this manuscript regarding ADC. I have the following comments for the authors -
1. I found redundant information regarding FDA-approved ADCs as this has been already published in recent review articles, such as ref. 148 (mentioned in this paper).
2. Following recent review articles are missing -
1. Potential of antibody–drug conjugates (ADCs) for cancer therapy. Hasan et al., 2022
2. Antibody drug conjugate: the “biological missile” for targeted cancer therapy. Zhang et al., 2022
3. There are many antibody-drug conjugates mentioned in table-3 but only a few have been discussed. Moreover, this list is not comprehensive as there are more ADCs in clinical development like AGS67E, AGS-62P1, MEDI4276, etc.
In my opinion including the following topics in this manuscript would make it more interesting -
1. Application of different computational techniques in addressing ADC challenges
2. Neoantigen-targeted cancer vaccines
Round 2
Reviewer 1 Report
In my opinion, there are many reviews on the same topic. Even though the authors have included some more information, I can not see a key difference. When a reader starts to read this review will not be motivated to continue since they will read same than in others reviews
Reviewer 3 Report
Now, this manuscript looks much better than the previous version and I am thankful to the authors for considering my suggestions.